

# Precipitation characteristics and associated weather conditions on the eastern slopes of the Rocky Mountains during March-April 2015

Julie M. Thériault[1], Ida Hung[2], Paul Vaquer[1], Ronald E. Stewart[2], and John Pomeroy[3]

[1]Univesrité du Québec à Montréal
[2]University of Manitoba
[3]University of Saskatchewan

**Correspondence:** Julie M. Thériault (theriault.julie@uqam.ca)

**Abstract.** Precipitation events that bring rain and snow to the Banff/Calgary area of Alberta are a critical aspect of the region's water cycle and can lead to major flooding events such as the June 2013 event that was the second most costly natural disaster in Canadian history. Because no special atmospheric oriented observations of these events have been made, a field experiment was conducted in March and April 2015 in Kananaskis, Alberta to begin to fill this gap. The goal was to characterize and better understand the formation of the precipitation at the surface during spring 2015 at a specific location in the Kananaskis Valley. Within the experiment, detailed measurements of precipitation and weather conditions were obtained, a vertically pointing Doppler radar was deployed, and soundings were launched. Although 17 precipitation events occurred, this period was associated with much less precipitation than normal (-25%), above-normal temperatures (+2.5°C) and below-normal relative humidity values (-2%). Of the 133 hours of observed precipitation, solid precipitation occurred 71% of the time, mixed precipitation occurred 9% and rain occurred 20%. An analysis of 17,504 images of precipitation particles showed that a wide variety of crystals and aggregates occurred and approximately 63% were rimed; this was largely independent of whether flows aloft were upslope (easterly) or downslope (westerly). In the often sub-saturated surface conditions, hydrometeors containing ice occurred at temperatures as high as 9°C. Radar structures aloft in association with precipitation were highly variable with reflectivity sometimes > 30 dBZe and vertical motions of particles sometimes reached ∼ 1 m/s upward within ascending air masses. Precipitation was formed in this region within cloud fields sometimes having variable structures and within which supercooled water at least sometimes existed to produce accreted particles massive enough to reach the surface through the generally dry sub-cloud region.

## 1 Introduction

The eastern slopes of the Canadian Rocky Mountains are prone to precipitation extremes. The region has a continental climate subject to extremes that fluctuate from severe drought to extensive flooding. An example is the flooding in June 2013, which was one of the most catastrophic events in Canadian history (Pomeroy et al., 2016; Liu et al., 2016; Kochtubajda et al., 2016). Previous floods such as in 2005 (Flesch and Reuter, 2011; Shook, 2016) have rivalled the 2013 event in terms of impact.

The climate of this area has been changing (DeBeer et al., 2016; Harder et al., 2015; Whitfield, 2014). Over the period 1950-2012, significant warming has occurred (2°C over the region) and it has been greatest during the winter (3.9°C). As for



precipitation over this period, it increased by 14% over the entire region although no trend in winter precipitation has been observed over southern Alberta. Changes are anticipated in the future climate of this region. All projections indicate continual warming but differ widely in regards to precipitation (see, for example, IPCC, 2013).

It is critical that precipitation be well understood, including its phase, because of its impact in this region and its changing
climate. Catastrophic events such as the 2013 Alberta flooding arose in part because most of the precipitation fell as rain on mountainsides and this acted to melt the existing snowpack and accentuate runoff (Pomeroy et al., 2016).

Given the importance of precipitation, its trends and its future occurrence, more attention needs to be placed on improving our understanding of the atmospheric conditions associated with the type of precipitation reaching the surface during the months of March and April in this area. There is nonetheless a gap in appreciating the atmospheric conditions leading to
precipitation as well as the characteristics of the precipitation itself. A field experiment was carried out in March-April 2015 in the Kananaskis Valley (Figure 1) to obtain critical information such as particle characteristics at the surface as well as radar reflectivity and Doppler velocity aloft in support of this need.

By utilizing this unique information, this study aims to better understand the precipitation characteristics and associated atmospheric driving mechanisms on the eastern slopes of the Canadian Rockies. Some of the specific scientific issues are placing
the observing period into a longer-term context, quantifying the temporal variability of precipitation (and its detailed features) at the surface in relation to conditions aloft, and understanding the roles of accretion and sublimation on the precipitation reaching the surface.

The manuscript is organized as follows. Section 2 describes the field project and the instrumentation deployed. Section 3 describes the events documented and specific case studies. Section 4 focuses on the characteristics of the precipitation and
associated atmospheric conditions and the precipitation processes are presented in Section 5. Section 6 places the results into perspective by comparing its findings with other studies across Canada. Section 6 provides the conclusions.

## 2   Overview of the field experiment

Precipitation events that occur in the Banff/Calgary area are a critical aspect of the region's water cycle and they can sometimes lead to major disasters such as the 2013 flooding. Such events can bring rain, snow or both to the area but no atmospheric-
oriented special observations (beyond those made with operational networks) have been carried out until the March-April 2015 experiment. Note that the Foothills Orographic Precipitation Experiment (Smith, 2008) was conducted between Calgary and Edmonton but characterized precipitation amounts and temperature at the surface on a transect perpendicular to the foothills. A summary of the precipitation events and operating instruments is provided in Table 1.

Multiple weather instruments were deployed in the Kananaskis Valley to study the precipitation characteristics and asso-
ciated weather conditions. The exact location of the measurements is the Kananaskis Emergency Services Centre (KES) site (Figure 1a) at an elevation of 1445 m and it is surrounded by topography up to 3010 m MSL (Figure 1b).

To characterize the precipitation at the surface, two instruments were deployed as well as a microphotography system. A Geonor weighing precipitation gauge, placed in a single Alter shield, was used to measure the amount of water equivalent



precipitation. Precipitation particles, sizes and fall speeds were measured using an OTT Parsivel optical disdrometer (Battaglia et al., 2010). Detailed photographs and observations of solid precipitation were also made. The method closely follows that described in Gibson and Stewart (2007) and used in other studies such as Henson et al. (2011) and Thériault et al. (2014). The microphotography utilized a 10.2 megapixel Nikon D80 digital SLR camera equipped with a 60 mm macro lens set at

its shortest focus and a mounted portable flash. The snowflakes were collected on a black velvet-covered pad that was placed outside for periods that ranged from 10 s up to several minutes, based on the precipitation intensity. The pad was then taken into a ventilated tent to be photographed. Approximately 9 images of the collection were taken in a systematic manner.

Manual weather observations were also made to complement the microphotography and to compare with other automatic measurements. Observations were generally recorded at 10 min intervals of precipitation conditions, cloud cover, temperature,

relative humidity, wind speed, wind direction and surface pressure; comments regarding conditions at the site and particles observed on the vehicle or surface prior to melting were also made. Several events contained precipitation particles that began to melt on contact with the velvet pad or the ground surface and microphotography was not feasible in these situations. Manual observations were recorded in these instances to characterize particle sizes and degree of riming.

Two weather stations (41 m apart) were utilized. Environment and Climate Change Canada (ECCC) provided one and the

other was the onsite Kananaskis Auto Boundary station (KBA) operated by Alberta Agriculture and Forestry (AAF). The ECCC station recorded 5 min data from 2140 UTC 17 March 2015 to 1705 UTC 23 March 2015 and the sampling frequency was increased to 1 min from 2057 UTC 24 March 2015 to 1621 UTC 30 April. The KBA station has been operating since 2005 (relative humidity since 2012) and records data at the top of each hour based on the average from the previous 59 min. Precipitation amounts are only available during the summer season. Because the precipitation gauge was not operational

throughout the project, the precipitation accumulation at Hay Meadow located in Marmot Creek was used. This station is located in Marmot Creek Research Basin (Pomeroy et al., 2012), 2.1 km from our site.

Both stations (KBA and ECCC) measured a standard suite of state variables (temperature, relative humidity, wind speed, wind direction, and surface pressure). Wind speed and direction sensors were measured at 3 m AGL (ECCC) and 10 m AGL (KBA) and temperature and humidity were measured at 0.9 m AGL (ECCC) and 2 m AGL (KBA).

The vertical structure of the atmosphere over KES was characterized using a vertically pointing Micro Rain Radar (MRR2). This instrument measures profiles of radar reflectivity and Doppler velocity (Klugmann et al., 1996). The vertical resolution of the MRR2 was 200 m and information up to 6200 m AGL is available.

The vertical temperature and moisture profiles were measured at 3 h intervals during each event by launching radiosondes. A total of 38 Vaisala radiosondes were launched although only 8 were released while microphotography observations were

being made. Note that the wind-determining capability did not work properly so no wind information is available.





## 3 Documented weather events

### 3.1 Overview of weather events

The March-April 2015 experimental period was anomalous in regards to temperature, precipitation and relative humidity. Temperature data obtained from the KBA station indicated an average of 2.6°C (from March-April 2015) which was 2.5°C
warmer than average (0.1°C from March-April 2006 to 2016). The average relative humidity was 63% (from March-April 2015), only 2% lower than the average (65% from March-April 2006 to 2016). The highest temperature recorded during the field campaign was 23.7°C on 2000 UTC 28 April 2015 and the lowest temperature was on 1300 UTC 17 March 2015 of -11.8°C. Precipitation collected at the Hay Meadow station during the study period (73 mm from March-April 2015) was 25% lower than the average (97 mm from March-April 2006 to 2016).

As summarized in Table 1, 17 precipitation events of snow (6 events), rain (2 events), and mixed precipitation (9 events) were documented. Note that no correlation has been found between the timing of precipitation events and the diurnal cycle. Overall, 133 h of precipitation was observed and documented. Of these, 71% were solid precipitation, 9% were mixed precipitation, and 20% were rain. The relationships among the precipitation types observed as well as the temperature and relative humidity are shown in Figure 2. Only precipitation associated with $T > 0°C$ is shown to highlight the presence of snow at positive values
and low values of relative humidity. Those at high temperatures maybe be water coating when the wet-bulb temperature is > 0°C.

The duration of each event varied and temperatures were often near or above 0°C. The longest period of continuous precipitation was 21 h on 15-16 March 2015 and the shortest was 2 h on 17 April 2015. Temperatures were > 0°C for 68% of the time when precipitation was observed. The warmest average event of 12.4°C occurred on 17 April 2015, and the coldest
average event of -0.5°C occurred on 4-5 April 2015 as well as on 6 April 2015.

Precipitation in this region is generally formed through orographic lifting associated with an easterly flow over the lee side of the Rocky Mountains. This was, however, not always the case during March and April 2015. Most of the precipitation events (11/17) were associated with a westerly flow aloft (3 km ASL) over the KES site. These were determined using the GEM-LAM (Global Environmental Multiscale-Limited Area Model simulations at 2.5 km resolution (Milbrandt et al., 2016). Examples of
the synoptic scale situation along with the MRR echoes at KES are presented below.

### 3.2 Large-scale weather conditions

This section will compare the synoptic-scale weather conditions that produced precipitation in the Kananaskis Valley during March and April 2015 based on flow field aloft. The case studies chosen occurred on 31 March 2015 for the westerly flow event and 4-5 April 2015 for the easterly flow one. An overview is given of the synoptic scale features as well as the MRR2
and surface observations collected during these events.

The analyses at 250 hPa and at sea level are shown in Figure 3. The westerly flow events were generally associated with an upper-level through located over B.C., which produced large-scale increasing cyclonic vorticity advection with height favourable for upward motion in the vicinity of KES (Figure 3a). This was supported by the presence of a surface low-pressure



system eastern Alberta (Figure 3c). In contrast, the upper-level trough is located farther west for the westerly flow. This led to a relative weak ridge near the border northern Alberta (Figure 3b), which produced subsidence and a high-pressure system and ridge at the surface over western Canada and a weak low-pressure system just south of the Canada-U.S. border (Figure 3d). These were the typical large-scale setups leading to the all the westerly and easterly flow field events, respectively.

To illustrate the atmospheric conditions aloft during both westerly and easterly flow events, examples of atmospheric soundings during each event are shown in Figure 4. The westerly flow events were generally dry (Figure 4a) because it would lead to downslope flow and the adiabatic heating would produce a dry layer near the surface. This easterly flow produced upslope conditions with adiabatic cooling, which led to saturation at a similar elevation to that of the barrier east of Kananaskis (Figure 4b).

## 3.3  Local Features Associated with Westerly and Easterly Flow Aloft

The evolution of precipitation at the surface varied systematically between the two flow regimes and the atmospheric conditions at KES are described and compared. The same two illustrative events from Section 3b are utilized. A representative westerly flow event occurred on 31 March to 1 April 2015 (Figure 5) and a representative easterly flow event occurred on 4-5 April 2015 (Figure 6). The data analysis is described in Appendix A.

First, the radar signal extended to more than 4 km AGL during the westerly flow event (Figure 5a, b) compared to only 2-4 km AGL during the easterly flow events (Figure 6a, b). The vertical particle motion was characterized by higher values during westerly flow event as some particles were moving upward whereas this was not observed during the easterly event (Figure 5b, 6b). A bright band was observed near the surface during the westerly flow event (Figure 5a, just after 2200 UTC). This is consistent with the $> 0°C$ surface temperature (Figure 5c) leading to rain and mixed phase precipitation (Figure 5f). In contrast,

the easterly flow event was generally colder (Figure 6c) with snow and more precipitation accumulated at the surface (up to 2.5 mm in liquid equivalent, Figure 6d) compared to that in the westerly flow event (1.5 mm, Figure 6d). Both events were associated with generally calm surface conditions although weak winds ($< 2.5$ m/s) sometimes occurred during the westerly flow event (Figure 5e and 6e).

On 4-5 April 2015, mainly snow reached the surface (Figure 6f) and particles were photographed. Most of them (23 UTC 4

April 2015 to 0500 UTC 5 April 2015) were a combination of rimed columns, planes, combinations of column and plane as well as aggregates. Rimed irregular particles were also photographed as well as a few occurrences of snow pellets. These rimed particles were sometimes observed simultaneously with unrimed ones suggesting that they were formed on different levels in the cloud.

Unusual particles were identified in images on 5 April 2015 and these are centered twelve-branched dendrites (Figure 7).

Only 3 of these were found among all the particles analyzed during the experiment. As discussed by (Kikuchi and Uyeda, 1998), the presence of such particles implies saturated conditions aloft at -18°C to -13°C; riming indicates the presence of supercooled droplets at these temperatures or at higher ones. The fact that the particles were intact at the surface also implies that they must have fallen through somewhat quiescent conditions with few if any collisions with other particles, which is confirmed by the stable conditions (low spectral Doppler width values from the MRR2).

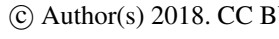



In general, the precipitation type evolution at the surface during the westerly flow events was rain changing into mixed precipitation. This occurred in 8 of the 11 westerly flow events. The height of the snow line, defined as the top of the layer with rain and mixed precipitation, varied between 1724 m ASL on 28 March 2015 to 1958 m on 17 April 2015. This was measured by the car-sonde along Fortress Mountain (cf Figure 1). On 31 March 2015, the height of the snowline was 1874 m, which
corresponded closely with the top of the bright band aloft (Figure 5a).

## 4   Precipitation characteristics at the surface

### 4.1   Overview of precipitation types at the surface

The ratio of each type of solid precipitation was diagnosed for all the events. The automatic method suggested that 24.1% of the precipitation occurrences were associated with rain (Figure 8) whereas the manual observations led to 16% rain as well as
6.3% mixed precipitation (cf Figure 2). Mixed precipitation can be very wet snow particles that can be diagnosed as raindrops. Given these generally consistent results, the diagnostic method is used to study the difference in the precipitation characteristics associated with varying flow field.

    The main difference in the precipitation types for the flow field regimes was the relative amount of rain and snow pellets reaching the surface. Rain mainly occurred (36.4% of the precipitation) during westerly flow events compared to 7.1% during
easterly flow events. In contrast, snow pellets commonly occurred during westerly flow event (47%) compared to easterly flow ones (5%). The fraction of densely rimed aggregates was nearly the same for both types of flow field.

    During all events, the box plots indicate that the median surface temperature was close to 1°C with a relative humidity of 82% (Figure 9). But warmer and drier conditions were associated with westerly flow (near 2°C and 81% relative humidity) events compared to those during easterly flow (0°C and 85%). The particle median diameter was 1.8 mm with a fall speed of
1.8 m/s for both types of events but the ranges differed. The westerly flow regime was associated with a wide range of fall speeds (0.5 - 4.5 m/s) whereas the range was much smaller (1 - 3 m/s) for the easterly flow regime. This is probably due to the higher fraction of rain during westerly events but also possibly due to more regions of upward air motions.

    The temperature, relative humidity, fall speed and diameter associated with the 4 main types of precipitation (rain, snow pellets, rimed snow and dry snow) for the flow field regimes are presented in Figure 10. The 2 categories of rimed aggregates
were combined and called rimed snow (RS). The size and fall speed varied with the type of event. The fall speed increased with the degree of riming, following the empirical relationship given in Appendix B. The particles with a lower degree of riming were associated with colder and more humid atmospheric conditions compared to the completely rimed particles, which is consistent with Figure 10. For example, snow pellets were associated with temperatures near 2°C and relative humidity of 70-80% whereas dry snow was associated with temperatures near -1°C and relative humidity around 80-90%.



## 4.2 Characteristics of solid precipitation

The detailed structure of single solid particles was investigated with the microphotography system. The largest single ice crystal (5 mm diameter) was observed on 4 April 2015 and the largest aggregate (24 mm) was observed on 14 April 2015. The number of each particle type is summarized in Figure 11a. The most common type of solid precipitation was irregular snow particles that Peterson et al. (1986) had described as being various snow crystals growing together at random. There was also a diverse occurrence of individual ice crystals as well as aggregates made up of different combinations.

Most of the unrimed particles with a discernible crystal habit were reported during one event (15-16 March 2015). During that event, 86% of the particles were unrimed. This event produced a variety of ice crystal (column and plane) and rimed particles. The degree of riming was also estimated on each particle (Figure 11b) using the classification proposed by Barthazy and Schefold (2006). Rimed particles were examined by eye in Adobe Photoshop and were placed into categories with respect to the degree of riming. Riming often occurred (63%) although its fraction varied substantially with particle type.

## 4.3 Occurrence of solid precipitation at temperatures $> 0°C$

Due to the dry atmospheric conditions in the region, 7 events produced ice crystals and solid precipitation particles at the surface when temperatures were above 0°C (cf Figure 2). Such instances had previously been found by Harder and Pomeroy (2013). The particles were identified by either manual observation or by microphotography. There were, however, many cases when the particles instantly melted once in contact with the surface or with an object, making it impossible to determine particle type.

The highest temperature at which particles were identified was on 25 April at 9.1°C with a relative humidity of 45%. This is consistent with the maximum temperature for snowfall observed over 7 years by Harder and Pomeroy (2013) at a higher elevation in the region. The average temperature during these instances was 3.8°C with a relative humidity of 57%. Therefore, a combination of dense (rimed) solid particles (Figure 11) and sub-saturated conditions near the surface (cf Figure 2) with low values of wet bulb temperature was responsible for the solid precipitation at the surface.

## 4.4 Idealized simulations

To demonstrate the impact of riming and sub-saturated conditions on the type of precipitation reaching the surface, idealized numerical simulations were conducted. A one-dimensional cloud model coupled to a microphysics scheme (Sankaré and Thériault, 2016) was used. The one-dimensional kinematic cloud model solves the mass divergence form of the continuity equation for water vapour and for all the other hydrometeor categories predicted by the bulk microphysics scheme. These are rain, snow, mixed precipitation such as almost completely melted particles and wet snow. The one-dimensional kinematic cloud model has been used in numerous studies (i.e. Milbrandt and Yau, 2005a, b; Thériault and Stewart, 2010; Sankaré and Thériault, 2016). For this purpose, the initial types of solid precipitation were unrimed snow (called bulk snow in (Sankaré and Thériault, 2016)) and snow pellets (called graupel in (Sankaré and Thériault, 2016)). The model is initialized with two vertical temperature and humidity profiles measured during the field campaign. These are 0500 UTC 18 April 2015 and 0800 UTC 18 April 2015, and




they are associated with rain/mixed precipitation and light snow, respectively. These entries are then interpolated over 100 vertical levels, which extends from the surface to above the melting layer (2 to 3 km). Since low precipitation rates were reported throughout March and April, the model was initialized with a rate of 1 mm/h.

The impact of the initial precipitation types aloft is shown in Figure 12. When snow is assumed to fall though the melting layer, mainly rain reached the surface 0500 UTC 18 April. In contrast, when snow pellets are assumed, a mixture of solid precipitation and mixed phase precipitation reached the surface. At this time, the surface temperature was 6.9°C and the depth of the warm layer ($T>0$°C) was approximately 900 m. The depth of the melting layer ($T_w>0$°C) is, however, only 500 m because the average relative humidity is 55%. Later during the event, light snow, made up of rimed aggregates, was reported at the surface. The observations are reproduced if snow pellets are initialized aloft. If unrimed snow is falling in the melting layer, 50% of precipitation would be rain, and $< 10\%$ of solid precipitation reaches the surface.

In summary, these idealized simulations demonstrated the impact of dry conditions and the type of solid precipitation aloft on the type of precipitation reaching the surface. A combination of dry conditions and completely rimed particles led to more solid precipitation at the surface. It is not possible to assess the reduction of the mass flux due to sublimation or evaporation because the calculations were conducted using the wet-bulb temperature. The wet-bulb temperature was computed, it was used at the air temperature in a saturated environment. This assumption allows computing the depth of the atmospheric layer where solid precipitation melts.

## 5   Precipitation processes aloft

The processes leading to the various types of precipitation at the surface were investigated using radar signals (MRR2) with respect to the flow field region. The Doppler velocity is defined to be positive if directed downwards.

The radar reflectivity, Doppler velocity and spectral width during the 2 types of flow field regime exhibited distinct patterns (Figure 13). Based on the reflectivity field, substantial precipitation (at least 5 dBZ) occurs over a deeper (near 1 km) layer aloft during westerly flow events compared to easterly flow. The Doppler velocity increased with descending height during westerly flow events, reaching 3 m/s near the surface. In contrast, during easterly flow events, the Doppler velocity was essentially constant with height (near 1 m/s). The higher values and wider range of the spectral width associated with the westerly flow events supports the surface observations that most of these precipitation events were associated with short and/or intermittent precipitation periods that could have been produced by local instability episodes. In contrast, the homogenous profile of low spectral width values explains the stratiform nature of precipitation caused by large-scale orographic lifting during easterly flow events.

The radar echoes were analyzed based on the types of precipitation and flow field (Figure 14). In general, each type of particle was associated with a wider range of Doppler velocity and spectral width values during westerly events. This general statement, however, cannot be made for the reflectivity patterns. Furthermore, the radar echoes differed for the same type of precipitation depending on the flow field. Large differences in the signal among precipitation types are evident during westerly flow events compared to easterly flow events. First, the precipitation layer leading to rain at the surface was at least 2 km deeper during





westerly flow events. Second, snow pellets had maximum reflectivity values during westerly flow events between 2000 and 4000 m as well as 1000 m AGL, and this maximum was not evident in the other types of solid precipitation and events. Within that layer, upward motion occurred, which is favourable for growth by accretion leading to completely rimed particles such as snow pellets. In contrast, upward motion was measured within a shallower layer when rimed snow reached the surface. In

summary, growth by accretion occurred in both types of event but at different elevations. Third, reflectivity values all decreased (except for downslope rain) with descending height just above the surface.

## 6    Prespectives

Most of the solid particles found during March-April 2015 were rimed or were mixed with unrimed particles even under dry surface conditions. These findings can be compared to the findings of some previous studies in other regions of Canada.

Studies of cold season precipitation features were carried out in the Whistler, British Columbia area in conjunction with the 2010 Winter Olympics. The degree of near-surface saturation varied during this period but was generally saturated or near-saturated (Isaac et al., 2014; Thériault et al., 2014). Associated precipitation was often rimed, up to 73% of the total number of particles.

A few studies have examined precipitation features in northern Canada. First, Burford and Stewart (1998) studied precipita-

tion at Inuvik and Tuktoyaktuk, Northwest Territories and found that it fell in light intensity or as trace amounts. They attributed this at least partially to the absence of accreted particles aloft; unrimed particles largely sublimated before reaching the surface. Second, precipitation events were examined at Fort Simpson, Northwest Territories (Hudak et al., 2004; Stewart et al., 2004). Precipitation particles were largely single crystals and aggregates; rimed particles were rare. The associated light precipitation rates were attributed at least partially to the lack of factors producing rimed particles aloft. Third, Henson et al. (2011) and

Fargey et al. (2014) examined precipitation in Iqaluit, Nunavut. Rimed particles, aggregates, and snow pellets were very common even during light precipitation events. It was suggested that the development of rimed and large particles increased their likelihood of reaching the surface through the dry sub-cloud layer.

In summary, cold season precipitation, such as occurring at Kananaskis, has been examined in several locations and the importance of accretion varied. At Kananaskis, this precipitation was associated with many rimed and unrimed particles. In

contrast, precipitation was observed or inferred to occur with many rimed particles at Inuvik, Tuktoyaktuk and Iqaluit and Whistler but not at Fort Simpson.

## 7    Conclusions

Precipitation on the eastern slopes of the Alberta Rockies is a critical issue as illustrated through numerous hydrological studies (Pomeroy et al., 2016). This article focused on atmospheric aspects of this precipitation utilizing a special dataset obtained in

March-April 2015 at the Kananaskis Emergency Services Centre. This particular period was associated with less precipitation, warmer conditions, and lower relative humidity than normal. The analysis of this information has led to several key points:



– Precipitation events were associated with both westerly (downslope) and easterly flow fields (downslope) aloft. It is generally considered that only easterly conditions are significant but 11 of the 17 events were associated with westerly flow.

– Even though the surface temperature was near 0°C throughout the study period, westerly flow events were generally warmer and drier at the surface than easterly flow ones. This is consistent with general descent in westerly flows and ascent in easterly ones. Higher occurrences of rain occurred during westerly flow events.

– The westerly flow events were generally associated with deep precipitation layers as well as more instability. The substantial precipitation layer (5 dBZ) was up to 1 km higher during those events but this also depended on the type of precipitation. For example, growth by accretion leading to snow pellets occurred at higher elevations during westerly flow events compared to easterly flow.

– Rain, snow and mixed phase precipitation all occurred at the surface and often within the same event.

– Different combinations of precipitation particles were often observed simultaneously, including rimed and unrimed particles, but the actual combinations varied dramatically, including within the same event.

– Solid precipitation particles reached the surface at temperatures up to 9°C and down to relative 45% humidity as previously reported by Harder and Pomeroy (2013).

– Approximately 60% of the solid precipitation particles were rimed with degree ranging from light to complete (such as snow pellets). Mixtures of rimed and unrimed particles had been observed in other regions under both near-saturated as well as dry conditions.

– All degrees of riming were associated with both westerly and easterly flows although the nature of the conditions aloft exhibited differences. For example, particles were sometimes moving upwards at higher elevations (> 4000 m AGL) during westerly flow events, which would be associated with supercooled droplet formation and an ideal environment for growth by accretion. Such instances did not occur during easterly flow events but these were moister with widespread ascent, which would have led to widespread droplet formation to allow accretion.

To conclude, it is recognized that the field experiment occurred during an anomalous period with no major event producing tremendous amounts of precipitation. Nonetheless, considerable insight has been gained into precipitation production even though a comprehensive understanding requires the examination of all types of situations.

*Data availability.* The dataset used to conduct this study is available through the Changing Cold Regions Network.



## Appendix A:  Data analysis

### A1    Analysis of precipitation microphotography

A detailed analysis of the microphotography of solid precipitation particles was carried out and this included a classification of the type of ice crystals, their degree of riming and the atmospheric conditions in which they occurred (Hung, 2017). Microphotography of particles was made during 7 precipitation events of the 17 precipitation events documented. Each image was analyzed using the Adobe Photoshop CS6 software. A point was placed on each ice crystals and solid precipitation particle using the Pen Tool to avoid duplication when being tallied and categorized. Overall, 1,183 images were collected and analyzed. After analyzing each image, 17,504 ice crystals and solid precipitation particles were identified with the most being collected on 4-5 April 2015.

Identified particles were organized into general categories adapted from the Kikuchi et al. (2013) classification of ice crystals and solid precipitation. For this study, irregular particles were placed into the irregular snow particle group. Particle types known as graupel-like snow and graupel particles from Kikuchi et al. (2013) were placed into a group referred to as the snow pellets category, created for this study for organizational and simplification purposes. Twelve categories were created, which have six types of crystal habits and whether they were rimed or not. Figure 15 illustrates the categories into which the particles were placed. For example, a rimed dendrite was placed into the category called rimed plane crystal group.

### A2    Diagnostic of precipitation types using the optical disdrometer

The type of precipitation was also diagnosed automatically using the optical disdrometer (Battaglia et al., 2010) based on the method proposed by Ishizaka et al. (2013). This technique takes into account the momentum of the particle. Further details are given in Vaquer (2017). To determine the particle type, the centre of mass flux ($CMF$) is computed and defined as

$$CMF = \sum_i fr / \sum_i r \qquad (A1)$$

where $r$ is a vector that indicates the location of each particle in the fall speed-diameter diagram, $f$ is the flux defined as $f = mv$, where m is the mass of the particle and v is the fall speed. Examples are shown in Figure 16a and c.

The fall speed-diameter and the mass-diameter relationships used to classify the main type of precipitation, based on the degree of riming, are given in Table 2. The main types of precipitation considered differ from those shown in Figure 16, which are snow pellet ($SP$), densely rimed aggregates ($HS$), rimed aggregates ($RA$), dry snow ($DS$) and dendrites ($DE$). These types reflect different degrees of riming rather than the specific crystal habit that generally falls at around 1 m/s for any diameter. To compare the microphotography analysis and the diagnostic of precipitation type from optical sensor information, densely rimed aggregates and rimed aggregates were combined to a single category called rimed snow ($R$S). See Appendix B for a comparison with the manual observations and the automatic diagnostic. All the particles with a diameter < 1 mm falling at < 2 m/s were neglected to allow the calculation of CMF (Ishizaka et al., 2013). Note that Yuter et al. (2006) also neglected the lower diameter values.




As an example, Figure 16 shows the recorded fall speed-diameter during 5 min, the computed $CFM$ and the associated solid precipitation observed at that time. Figures 16a and c are for densely rimed aggregates and snow pellets, respectively. The sizes of particles estimated from the microphotographs corresponded well with the optical disdrometer measurements. In particular, the densely rimed aggregate had a diameter about 3 mm (Figure 16b) and a more rimed particle had a diameter about
2 mm (Figure 16d).

## A3    Atmospheric conditions aloft

Precipitation in this region is generally formed through orographic lifting associated with an easterly flow over the lee side of the Rocky Mountains. This was, however, not always the case during March and April 2015. Most of the precipitation events (11/17) were associated with a westerly flow aloft at the KES site (Table 1).

To classify the weather events based on the flow regime, the GEM-LAM (Milbrandt et al., 2016, Global Environmental Multiscale-Limited Area Model) model simulations at 2.5 km resolution were used. A 6 x 6 grid point was used at a frequency output of 1 h resolution. More details is given in Vaquer (2017).

To correlate the surface weather conditions with the atmospheric conditions aloft, the data collected by the MRR2 were analyzed. The method developed by Maahn and Kollias (2012) for snow was used to process the data and to obtain the
equivalent reflectivity ($Ze$), Doppler velocity ($W$) and spectral width ($\sigma$). The types of precipitation at the surface diagnosed with the optical disdrometer were then correlated with the MRR2 signals and the wind direction at 2000 m above sea level. Only the events with operational optical disdrometer, MRR2 and ECCC weather station (Table 1) were used for this investigation. We used the diagnosed precipitation type instead of the microphotography to consider all precipitation types and events. A comparison between the disdrometer diagnostic and the particle microphotography was performed when solid precipitation
occurred. The results are summarized in Appendix B.

## Appendix B: Comparison of precipitation type and particles

A comparison of the precipitation observed or inferred by the optical disdrometer, manual observations, and microphotography was carried out (Table 3). Although the optical disdrometer only considered specific precipitation types, the data are generally comparable to the manual observations. However, the optical disdrometer inferred rain on several instances during the April
4-5 and April 11-12 events whereas snow was reported in the manual observations. As well, rain was inferred on April 25-26 when mixed precipitation was reported in manual observations.

The twelve categories identified in the microphotography observations were compared to the optical disdrometer precipitation types. Precipitation categories were generally comparable when both observations identified rimed and aggregated particles but not for rain events. When snow pellet, densely rimed aggregates, or rimed aggregates were inferred by the opti-
30    cal disdrometer, microphotography identified rimed particles, rimed aggregates, or snow pellets. In contrast, when the optical disdrometer inferred rain mixed with several types of ice particles, it correlated with manual observations but not always with





manual photography. Overall, the optical disdrometer is reasonably comparable with both the manual and microphotography observations.

*Competing interests.*  Author, John Pomeroy, is a guest editor of the special issue, Understanding and predicting Earth system and hydrological change in cold regions.

5  *Acknowledgements.*  The authors would like to thank the many people who contributed to the successful field experiment. These are Juris Almonte, Stephen Berg, Émilie Bresson, Mélissa Cholette, Dominic Matte, Émilie Poirier, Housseyni Sankare, Bruce Cole, Craig Smith, May Guan, Angus Duncan, Scott Landolt, Al Jachick and Roy Rasmussen as well as the Biogeosciences Institute of the University of Calgary and the Kananaskis Emergency Services Centre firefighters. The research was supported by the Changing Cold Regions Network (CCRN), which was funded by the Natural Sciences and Engineering Research Council of Canada (NSERC) and by NSERC Discovery grants of J.
10  Thériault and R. Stewart.





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



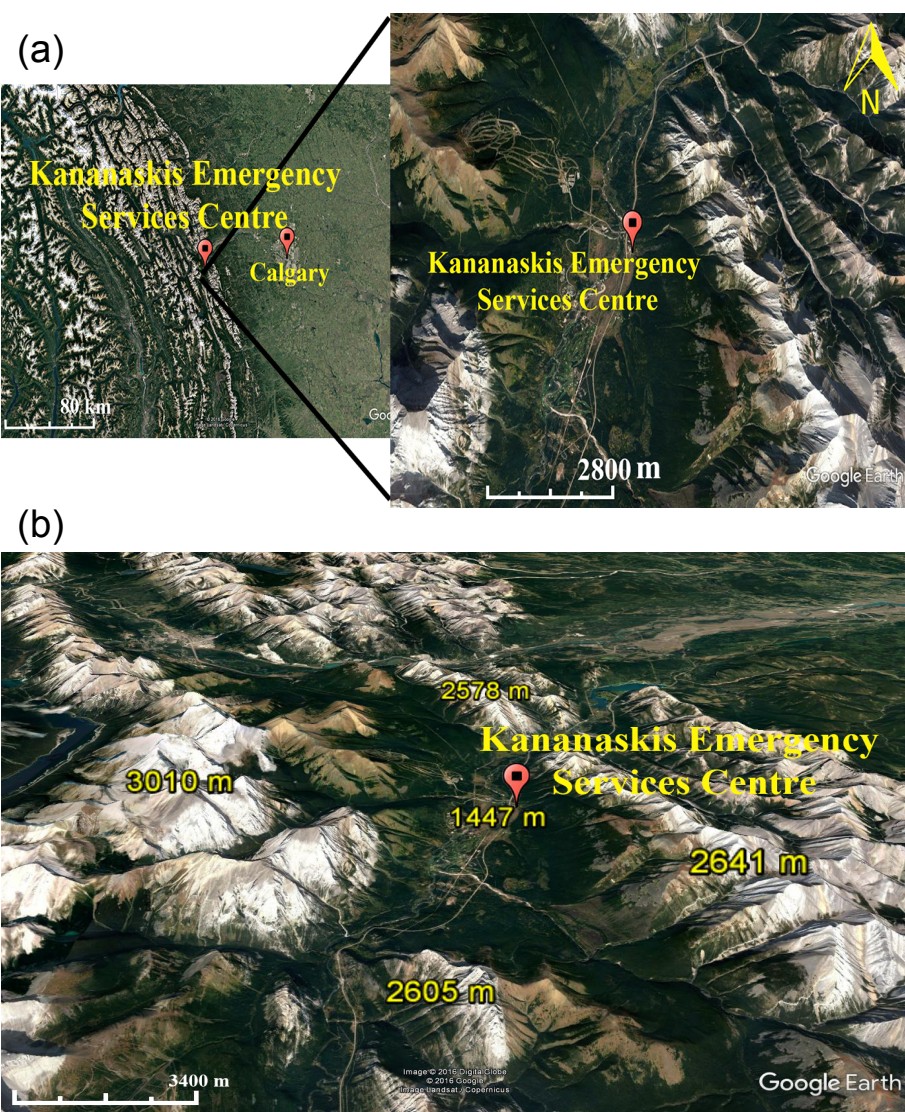

**Figure 1.** (a) The Kananaskis Emergency Services Centre site is located in the lee of the Rocky Mountains at 50.9° latitude, -115.1° longitude and (b) is surrounded by topography up to 3010 m MSL. Google Earth. 23 September 2012. Retrieved on 16 January 2017.




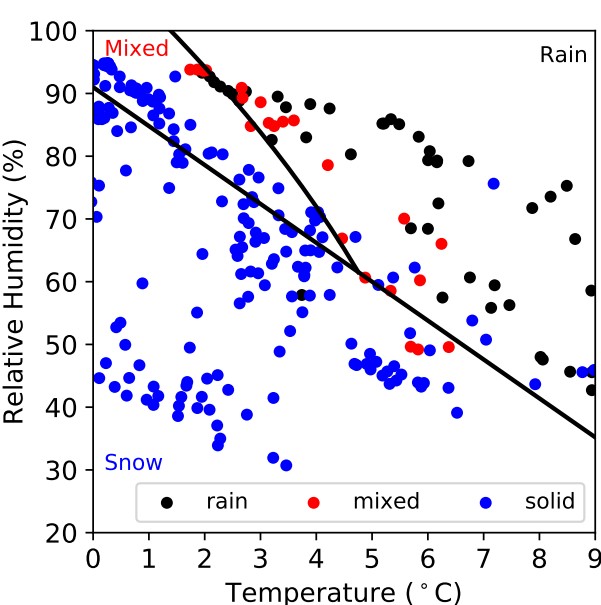

**Figure 2.** The types of precipitation with respect to the temperature and relative humidity for all the events over the March-April 2015 period using the manual observations and the weather station. The background threshold lines are from Matsuo et al. (1981).




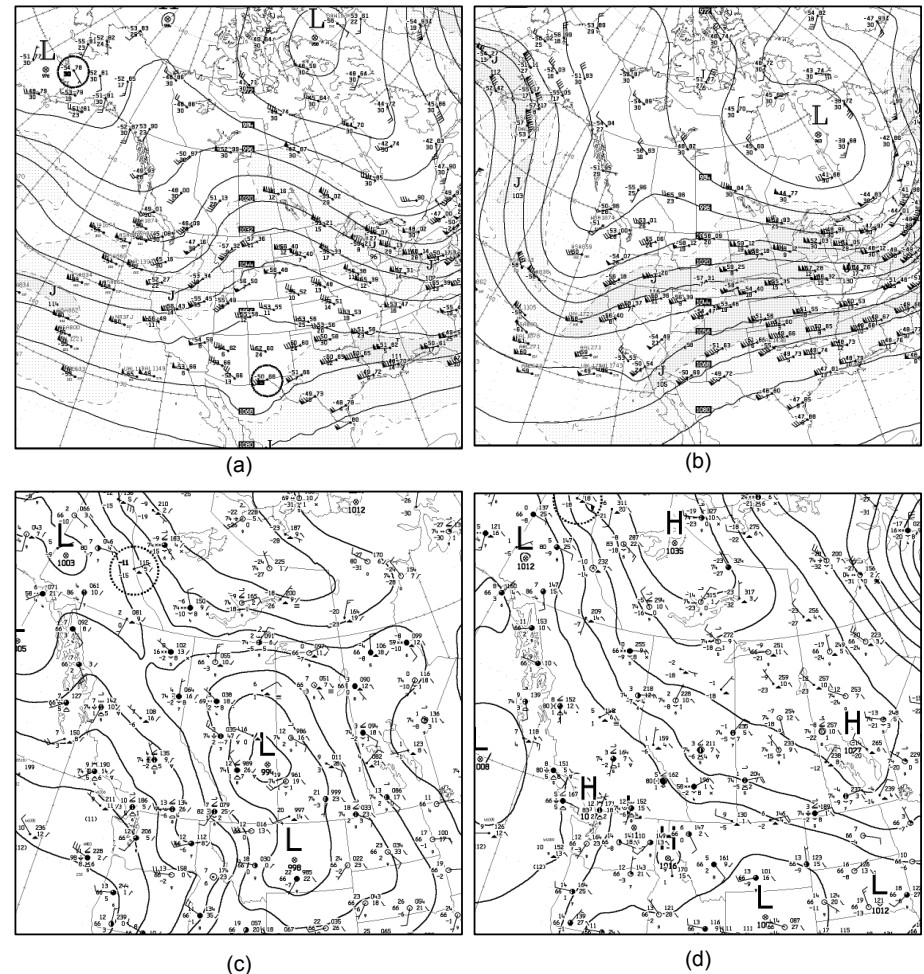

**Figure 3.** (a) and (b) are the 250 hPa analyses of the geopotential heights and wind fields. (c) and (d) are the analyses of the surface weather conditions at sea level. (a) and (c) are at 0000 UTC 1 April 2015, which corresponds to a westerly flow event and (b) and (d) are at 0000 UTC 5 April 2015, which corresponds to a easterly flow event.





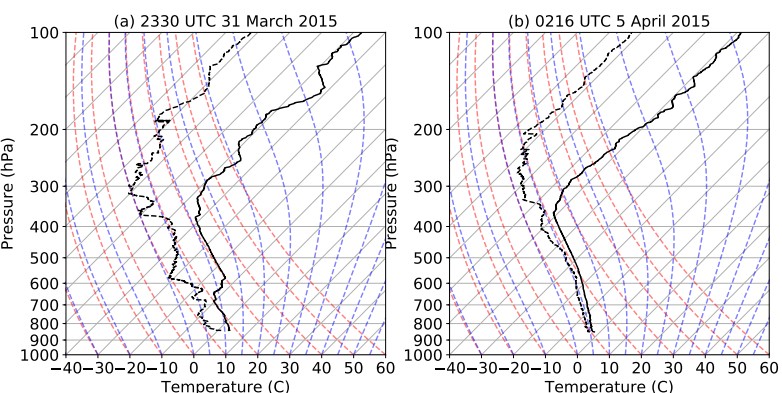

**Figure 4.** The soundings launched at (a) 2330 UTC 31 March 2015 (westerly flow event) and (b) 0216 UTC 5 April 2015 (easterly flow event).



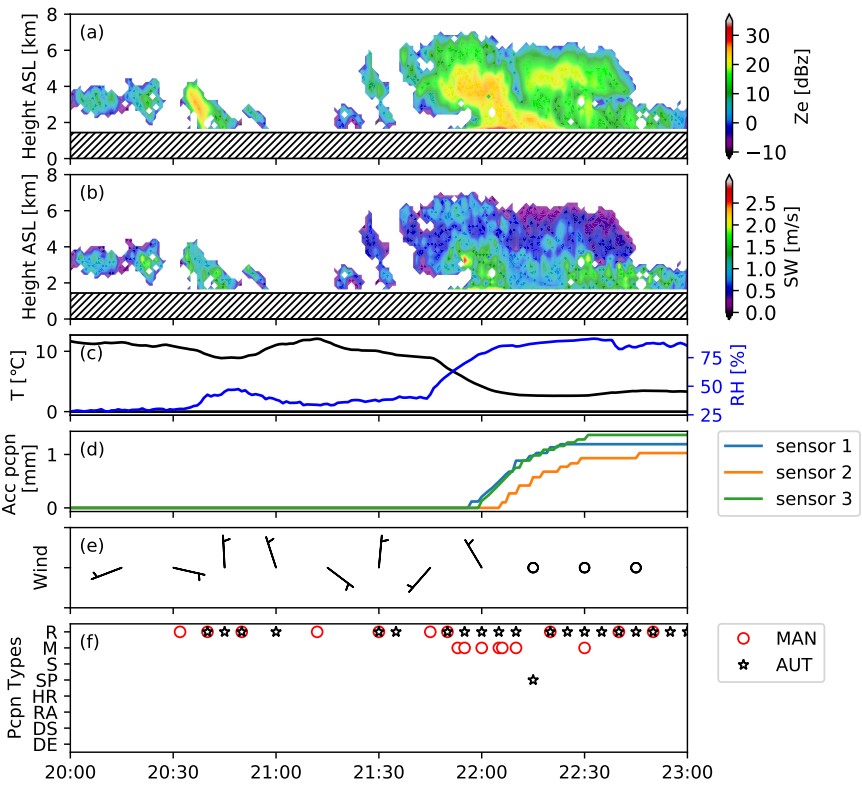

**Figure 5.** Time evolution (UTC) on 31 March 2015 of (a) reflectivity and (b) spectral width fields retrieved from the Micro Rain Radar 2 (MRR2) for the westerly flow field event that occurred on 31 March 2015. (c) is the surface temperature and relative humidity; (d) is the wind speeds and direction represented by the wind barbs; (e) is the accumulated precipitation at the surface (KES) and (f) is the manual observation of precipitation types as well as the one diagnosed using the OTT Parsivel (Battaglia et al., 2010). The hatched surface on the MRR time series (a, b) indicates ground-level height. The MRR2 quality control was based on Maahn and Kollias (2012). See Appendix A for the details on the precipitation type diagnostic method.





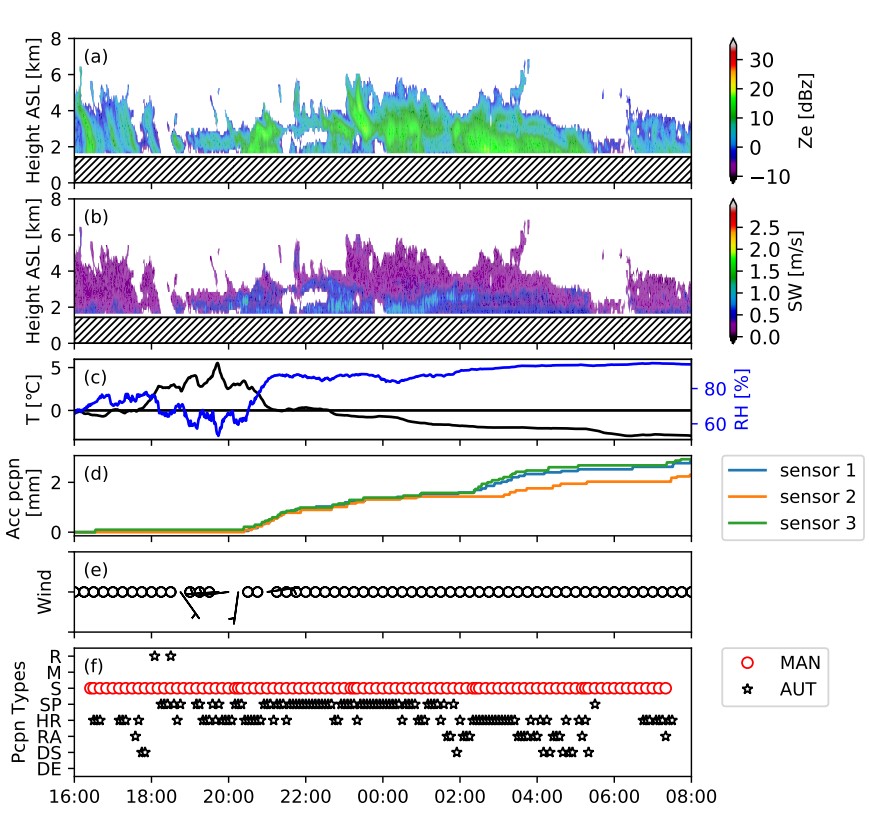

**Figure 6.** Same as Figure 5 but for an easterly flow event that occurred on 4-5 April 2015.

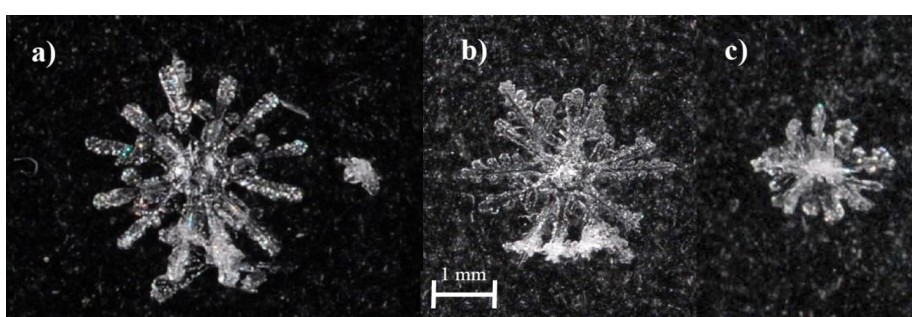

**Figure 7.** Twelve-branched dendrites were identified on 5 April 2015 at (a) 0334 UTC, (b) 0341 UTC, and (c) 0512 UTC. The scale is the same for each particle. These particles ranged in size from approximately 2 mm to 4 mm across and were lightly rimed. The average temperature when the particles were photographed at the surface was -2°C with an average relative humidity of 93%.





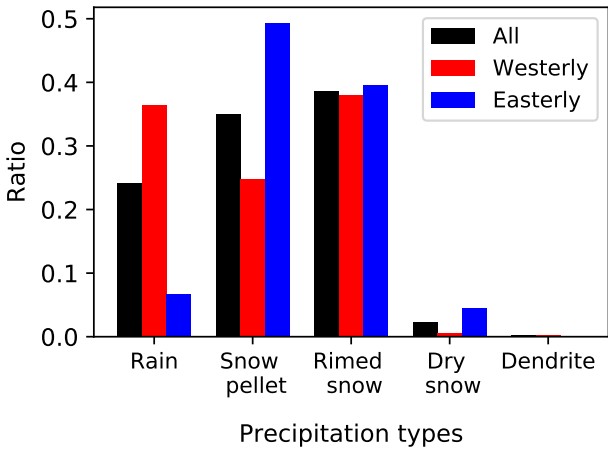

**Figure 8.** The fraction (%) of precipitation and crystal types diagnosed with the Ishizaka et al. (2013) method using optical disdrometer information over the March-April 2015 period. A summary is provided for all events, westerly flow events and easterly flow events.





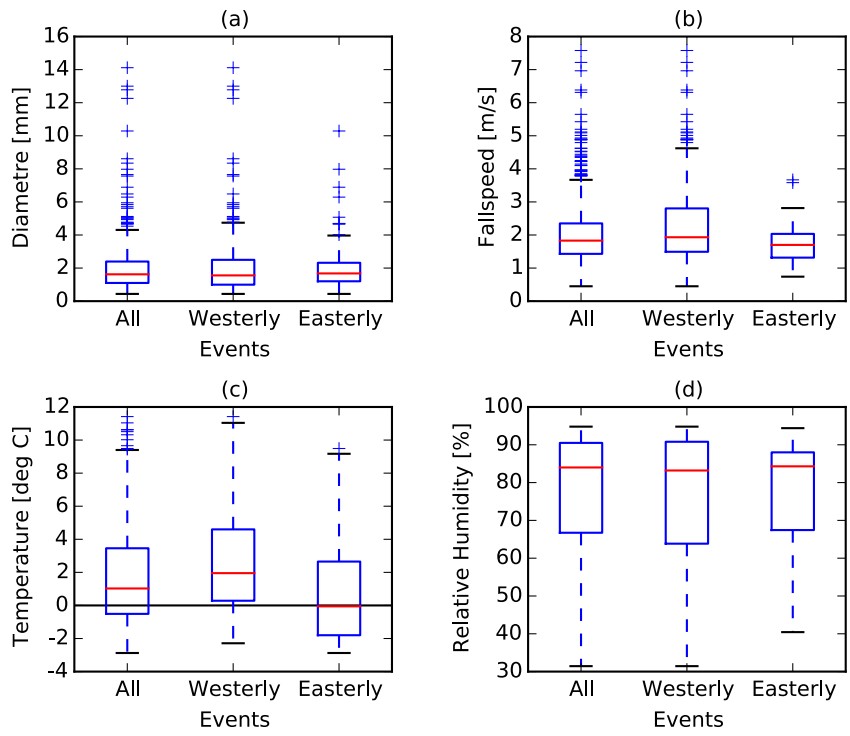

**Figure 9.** Summary over the March-April 2015 period of the (a) diameter, (b) fall speed, (c) temperature and (d) relative humidity represented by a box plots. The red line is the median, the box is between the 25th and 75th percentiles and the whiskers are the 5th and 95th percentiles.





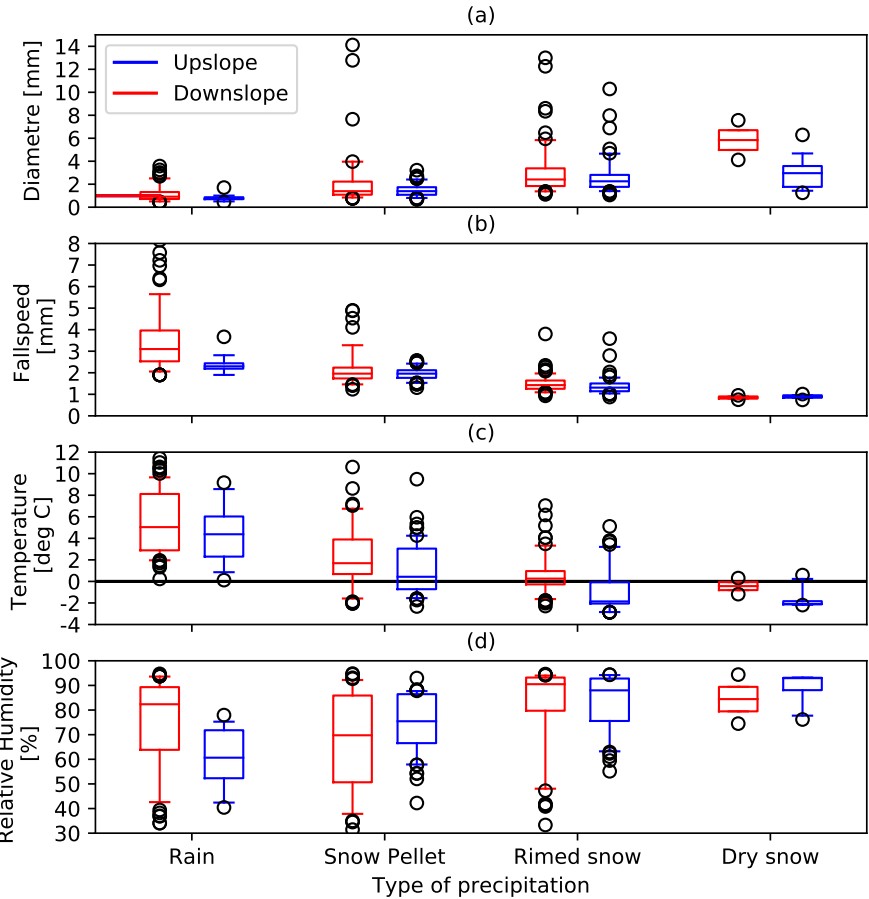

**Figure 10.** Comparison of the surface conditions associated with the 4 main types of precipitation over the March-April 2015 period. The main types are rain (R), snow pellets (SP), rimed (RS) and dry snow (DS) and these are shown occurring during westerly (red) and easterly (blue) flow events. (a) is the diameter, (b) the fall speed, (c) the air temperature and (d) the relative humidity. The red line is the median, the box is between the 25th and 75th percentiles and the whiskers are the 5th and 95th percentiles.



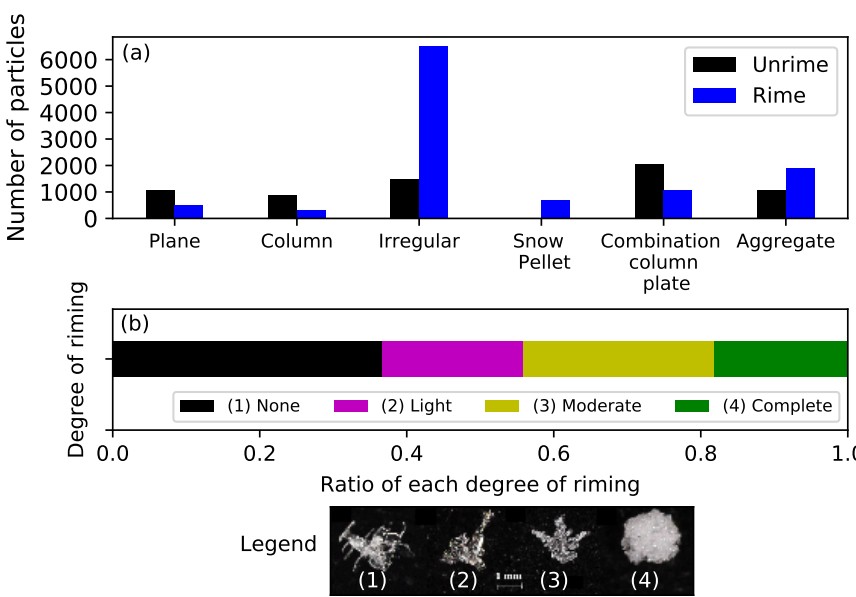

**Figure 11.** (a) The number of unrimed and rimed ice crystals and solid precipitation particles identified using the microphotography observations over the March-April 2015 period. For simplification purposes, the other solid precipitation category was omitted from the image since fewer than 5 of these particles were observed. (b) The fraction of particles with varying degrees of riming as identified in microphotography observations over the March-April 2015 period. The insert provides a visual reference for the different categories of riming.




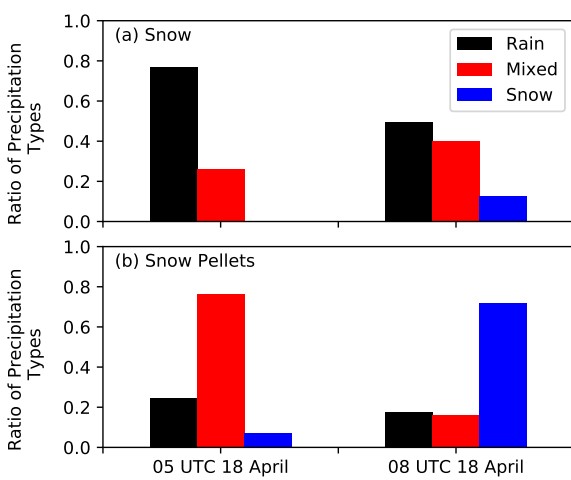

**Figure 12.** Ratio of precipitation produced at the surface by the one-dimension cloud model coupled to a bulk microphysics scheme (Sankaré and Thériault, 2016). The precipitation produced assuming (a) snow and (b) snow pellets falling through the melting layer ($T > 0°$C) associated with 2 soundings.




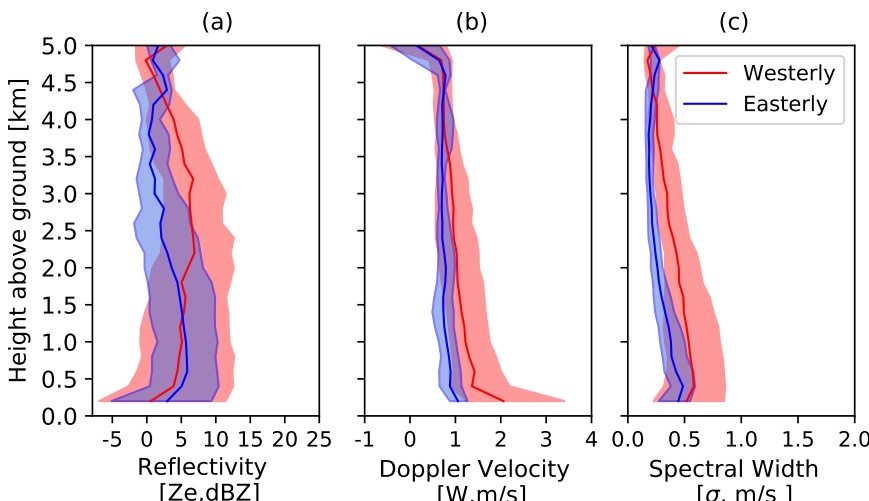

**Figure 13.** MRR2 profiles with height (AGL) associated with both westerly (downslope) and easterly (upslope) flow events over the March-April 2015 period. (a) is equivalent reflectivity, ($Ze$, dBZ), (b) is Doppler velocity ($W$, m/s) and (c) is spectral width ($\sigma$, m/s). The height is above the radar and data from the first 200 m are not used to eliminate surface-induced noise. The color shading is the region bound by the 25th and 75th percentiles. The median is the bold line.





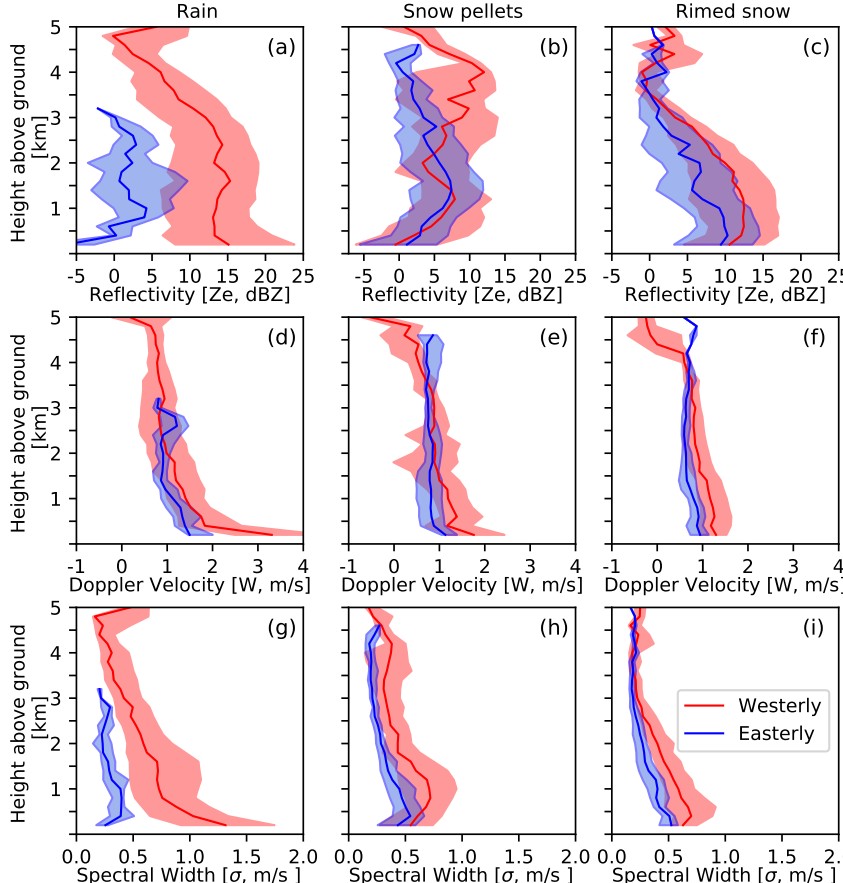

**Figure 14.** MRR2 profiles with height (AGL) associated with both westerly (downslope) and easterly (upslope) flow events over the March-April 2015 period for 3 types of precipitation: (a, d, g) rain, (b,e,h) snow pellets and (c, f, i) rimed snow. (a-d) are reflectivity, Ze (dBZ), (d-f) are Doppler velocity, W (m/s) and (g-i) are spectral width, $\sigma$ (m/s). The height is above the radar and data from the first 200 m are not used to eliminate surface-induced noise. The color shading is the region bound by the 25th and 75th percentiles. The median is the bold line.




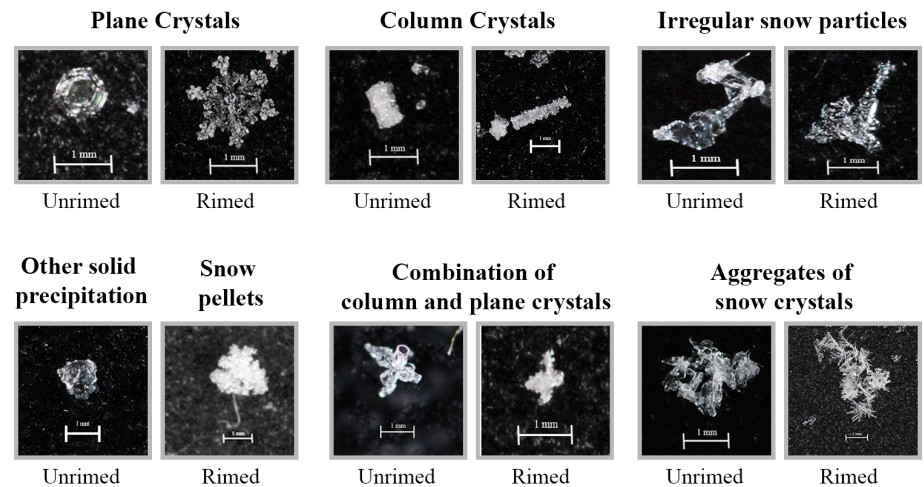

**Figure 15.** The 12 categories of ice particles utilized in the analysis of the microphotography images. The categories are adapted from Kikuchi et al. (2013).





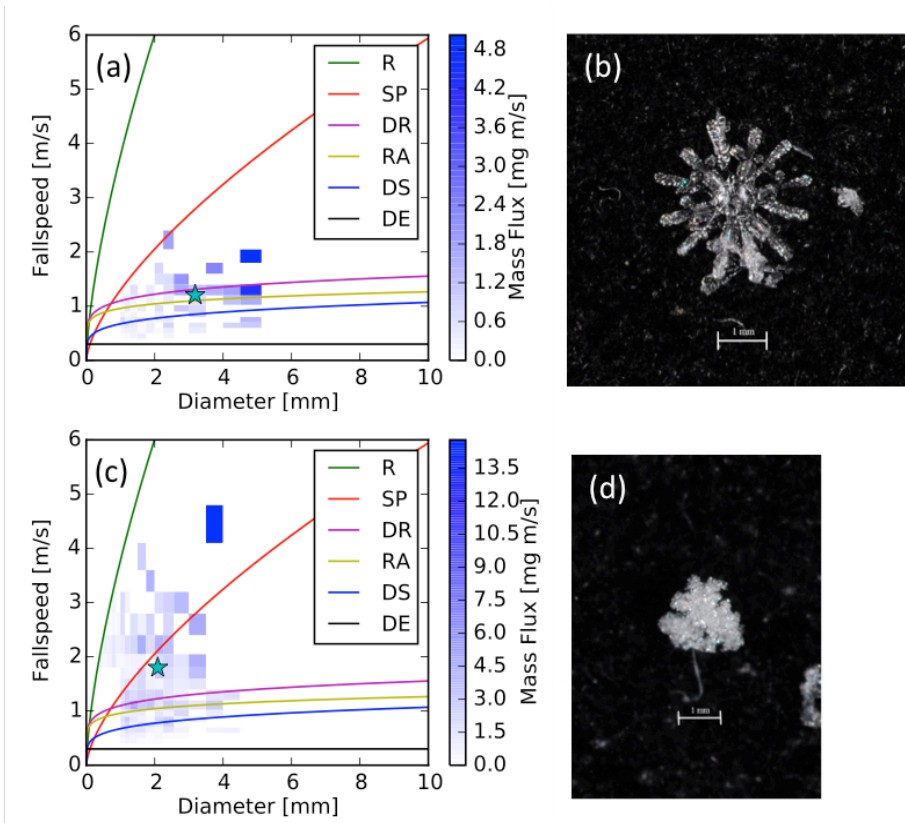

**Figure 16.** (a, c) The 5-min average of the fall speed-diameter distributions within which two types of snow have been diagnosed using the Ishizaka et al. (2013) method. (b,d) are the associated snowflake images, respectively. The green star indicates the $CMF$ associated with the main type of particles. (a, b) is associated with a rimed particles and (c, d) is associated with snow pellet. The abbreviations of particle types are defined in Table 2.





**Table 1.** A summary of the events and instrumentation over the March-April 2015 observing period at the KES site. For each event, the mean temperature ($\bar{T}$) and relative humidity ($\bar{RH}$) are given. The amount and types of precipitation at the surface were rain ($R$), mixed ($M$) and/or snow ($S$). The direction of the flow field aloft (*Type event*), which was either an easterly flow event ($E$) or a westerly event ($W$), is also indicated. A X indicates when each instrument was operating. These are the surface manual observations ($SMO$), the surface meteorological station ($SMS$), the optical disdrometer ($OD$), the Micro Rain Radar ($MRR2$), the microphotography setup ($MP$) and the sounding system ($SS$).

| Num | Event | $\bar{T}$ [°] | $\bar{RH}$ [%] | Amount pcpn* [mm] | Type pcpn | Duration [h] | Type event | SMO | SMS | OD | MRR2 | MP | SS |
|---|---|---|---|---|---|---|---|---|---|---|---|---|---|
| 1 | 15-16 March | | | 8.7 | S | 21 | E | X | | X | | X | |
| 2 | 21-22 March | | | 0.05 | R→M→S | 11 | W | X | X | X | | X | |
| 3 | 23 March | | | 0.08 | S→M | 3 | E | X | X | | X | | X |
| 4 | 28 March | 5.7 | 82 | 0.04 | R→M | 6 | W | X | X | X | X | | X |
| 5 | 30 March | 6.1 | 67 | 0.01 | R | 2 | W | X | X | | X | | X |
| 6 | 31 March | 7.0 | 58 | 0.03 | R→M | 3 | W | X | X | X | X | | X |
| 7 | 1-2 April | 0.8 | 54 | 0.73 | S | 9 | E | X | X | X | X | | X |
| 8 | 4-5 April | -0.5 | 83 | 1.73 | S | 15 | E | X | X | X | X | X | X |
| 9 | 6 April | -0.5 | 62 | 0.13 | S | 6 | W | X | X | X | X | | X |
| 10 | 11-12 April | 2 | 63 | 1.70 | S | 20 | W | X | X | X | X | X | X |
| 11 | 14-15 April | 3.2 | 76 | 3.11 | R→M→S | 11 | W | X | X | X | X | X | X |
| 12 | 17 April | 12.4 | 52 | 0.11 | R | 2 | W | X | X | X | X | | X |
| 13 | 18 April | 1.6 | 79 | 1.34 | R→M→S | 16.75 | W | X | X | X | X | X | X |
| 14 | 22-23 April | 9.9 | 42 | 0.04 | R→S | 7 | W | X | X | X | X | | X |
| 15 | 24-25 April | 6.3 | 49 | 0.34 | S | 11 | W | X | X | X | X | | X |
| 16 | 25-26 April | 3.4 | 65 | 0.14 | M→S | 6 | E | X | X | X | X | X | X |
| 17 | 29 April | 10.6 | 46 | 1.01 | R→M→S | 10 | W | X | X | X | X | | X |

*Data retrieved from Marmot Creek, Hay Meadow station data





**Table 2.** Fall speed-diameter, $v(D) = aD^b$ in m/s, and mass-diameter, $m(D) = cD^d$ in mg, used to diagnose the type of precipitation using the Ishizaka et al. (2013) method. The values are from Atlas and Ulbrich (1977), Locatelli and Hobbs (1974), Ishizaka (1995), Rasmussen et al. (1999) and Nakaya (1954), respectively. The diameter, $D$, is in mm.

| Particle type | Symbol | a [$10^{-3}$ mm$^{1-b}$/s] | b | c [mg mm$^{-d}$] | d |
|---|---|---|---|---|---|
| Rain | $R$ | 3.78 | 0.67 | 0.52 | 3 |
| Snow Pellets | $SP$ | 1.3 | 0.66 | 0.078 | 2.8 |
| Densely Rimed Aggregates | $DR$ | 1.1 | 0.57 | 0.094 | 1.9 |
| Rimed Aggregrates | $RA$ | 0.96 | 0.12 | 0.068 | 1.9 |
| Dry Snow | $DS$ | 0.107 | 0.2 | 0.089 | 2 |
| Dendrites | $DE$ | 0.3 | 0 | 0.0038 | 2 |



**Table 3.** A comparison of the optical disdrometer, manual, and microphotography observations over the March-April 2015 period. Events between 15 March 23 March 2015 were omitted due to missing data.

| Event | Optical disdrometer | Manual observations | Microphotography |
|---|---|---|---|
| 28 March | Rain | Rain | – |
| 31 March | Rain | Rain, snow | – |
| 1-2 April | Snow pellets, rimed, dry snow | Snow (rimed) | – |
| 4-5 April | Rain, snow pellets, rimed, dry snow | Snow (dendrites, rimed, aggregates) | Rimed irregular particles, rimed aggregates of snow crystals, rimed combination of column and plane |
| 11-12 April | Rain, snow pellets, rimed, dry snow, dendrites | Snow (aggregates, densely rimed) | Rimed irregular particles, snow pellets, rimed aggregates of snow crystals |
| 14-15 April | Rain, snow pellets, rimed | Rain, snow (snow pellets, rimed, aggregates) | Rimed irregular particles, rimed aggregates of snow crystals |
| 18 April | Rain, snow pellets, rimed, dry snow | Rain, snow (snow pellets, rimed) | Rimed irregular particles, rimed aggregates of snow crystals, rimed plane crystals |
| 22-23 April | Rain | Rain | – |
| 24-25 April | Rain, snow pellets, rimed | – | |
| 25-26 April | Rain, snow pellets, rimed | Mixed precipitation, snow (snow pellets) | Snow pellets, rimed irregular particles |
| 29 April | Rain | Rain, snow | – |