# Peer review of "Precipitation characteristics and associated weather conditions on the eastern slopes of the Rocky Mountains during March-April 2015"

_Hydrology and Earth System Sciences, 2018_

## Referee Comment (RC1) · Anonymous Referee #1 · 3 May 2018

Overview

The study's authors present a thorough and wide-ranging study of the precipitation characteristics during a filed campaign. The manuscript is well written and organized, and the subject material is interesting. The analyses are detailed and robust. The manuscript will, however, benefit by addressing several key concerns identified by the reviewer. My recommendation is publication once the concerns listed below have been appropriately addressed.

[Figure]

Main Concerns

1) The authors state the importance of major summertime floods in the region and that no dedicated atmospheric study of such events have been made. While true, this claim is problematic when framed in the context of this particular study, which concerns low-intensity precipitation events observed during the spring months. My suggestion would be for the authors to frame the context of this work as it pertains to winter and spring precipitation, and how the characteristics of the observed precipitation/hydrometeor might affect i) the snowpack in terms of SWE, and ii) the spring freshet. On a related note, one could easily justify the need for a similar study in June, when heavy precipitation events typically affect this region.

2) The authors correctly point out that the climate in this region is changing. However, they do not discuss how the climate has changed in the spring months (i.e., the time period when the observations were made). Also, no connection is made between the long-term trends in temperature and precipitation and the objectives of this study. If they spring climate has been warming and drying, for example, how does that relate to this particular research initiative? That link needs to be clearly stated and substantiated with references to the literature.

3) The reasoning behind the motivation provided on lines 7 through 9 is not clear to me. Mention of made of the importance of precipitation, trends and future occurrence, but no specific discussion on these points is made in the preceding text. The connection between these aspects and this research needs to be clearly laid out.

4) A major focus of this research is how precipitation differs between westerly and easterly flow regimes. In my opinion, while logical, this classification may be too simple, especially considering the location of the study site and change of wind direction with height. For example, oftentimes winds in this region veer markedly with height, so an upslope flow (i.e., easterly) in the low levels may transition to westerly or southwesterly near the barrier height. The absence of winds from the radiosonde soundings, or discussion of vertical wind profiles in general, make correctly differentiating between "true" westerly and easterly events problematic. This could have important implications for the interpretation of results and conclusions drawn in this study. I strongly suggest that the authors use either data from the HRDPS or a higher resolution reanalysis product (e.g., NARR) to generate time-height diagrams of winds above the study site. These data, even with their limitations, can be used to draw more nuanced interpretations of how the winds vary between events and how the vertical profiles of winds affect precipitation.

5) Another important consideration is the impact of the local orography on the winds, as this could have important implications on the flow relative to the orography, vertical motions and ultimately precipitation. For example, according to Fig. 1, the study site was located on the western flank of a pronounced ridge on the front range having heights exceeding 2500 m. Whereas an easterly flow would be upslope on the eastern flanks of this ridge, is not clear what happens after the air passes over the ridge-line. It could be that the easterly flow actually results in a localized area of downslope flow on the western flank of the ridge. Similarly, a large-scale westerly flow could become upslope on the western flank of the ridge, resulting in localized vertical lift.

6) Precipitation amounts—the manuscript would benefit from a brief discussion of the precipitation amounts observed for the different flow regimes. Also, are heavier amounts typically favoured by a particular flow regime, and if so, why?

7) The manuscript would also benefit from a discussion on the impact of hydrometeor drift, as non-negligible precipitation amounts can arise because of hydrometeor drift from upslope flow on windward slopes.

8) Data from a sonde mounted on a vehicle (i.e., car-sonde) is referred to in the text. Details concerning these data need to be provided—maybe as an Appendix? Also, reference in the text is made to data gathered whilst ascending Fortress Mountain; please indicate the route and location of Fortress Mountain in Fig. 1.

Technical Comments

1) Page 2, lines 13-14: Make sure that this pertains to spring or cold season only.

2) Page 2, lines 26-27: The reference to FOPEX seems to be an afterthought. Reasons why the findings from FOPEX are, or are not, relevant to this study need to be clearly substantiated. FOPEX also considered the impact of altitude on precipitation amounts.

3) Page 3, line 7: It is not clear what is meant by "...taken in a systematic manner". Please reword for clarity.

4) Page 3 lines 19 and 29: Given the large instrument suite, please make sure that it is clear what instrument is being spoken to in the text. On line 19 are the authors referring to the Geonor gauge? On line 29, what sondes were used? Vaisala RS92s?

5) Page 4, line 3: Is the mean observed RH of 63% really anomalous compared to the average of 65%? Is the difference even statistically significant?

6) Page 4, line 14: Referring to Figure 2, this figure could be another useful contribution from this paper. Although similar, there are noteworthy shifts in the boundaries for these data compared to those from Matsuo et al. There are data points to warrant shifting the demarcation lines between categories. This would be useful to others.

7) Page 4, lines 15-16: The discussion of the wet-bulb temperature comes out of the blue. The importance and relevance of this parameter needs to be introduced first. What do "low values of relative humidity" translate to in percent?

8) Page 4, line 31: This is not a big issue, but using 250-mb to describe synoptic systems is atypical. Was there any particular reason of using data at this level at not 250-mb? Also, it would also be interesting to look at data from 700-mb as this coincides closely to the height of the mountain barrier.

9) Page 5, line 4: That these synoptic setups corresponded with "all" of the westerly and easterly flow regimes seems too definitive. Unless this can be quantified, maybe

more conservative language is warranted.

10) Page 5, lines 6-8: This may be pedantic, but the others refer to "dry" several times. It would be more appropriate to say "drier" given that we are referring to precipitation events, albeit with unsaturated conditions below cloud base. Please check text throughout. Also, the wording used to describe the impact of the flow regime on the relative humidity makes it sound like conjecture. Maybe rewording would address this, although using in-situ data that substantiate the claims would be best.

11) Page 5, line 8: In Fig. 4b the profile looks near-saturated from just above the surface.

12) Figures 5 and 6: Panels c and d are swapped in the figure caption. No units are provided for the surface winds, and no explanation of what speeds the barbs represent. It is not clear what sensor1, sensor2 and sensor3 represent—please clarify. Is there a particular reason why accumulation estimates from sensor2 are lower than the other sensors for both events?

13) Section 3.3. It is not obvious to me why the authors chose to plot the spectral width in Figs. 5,6 and not the Doppler-derived vertical motions. Spectral width data are typically used as a proxy for turbulent motion. Sound reasoning needs to be provided for the choice of spectral width data when speaking to vertical motions of hydrometeors. The choice of spectral width is also confusing because on line 16 on page 5, reference is made to "the vertical particle motion", which suggests that the authors are referring to radial velocity data.

14) Page 6, lines 21-22: Regarding ". . .but also due to more regions of upward motions". It is not clear on which data this assertion is made. Please clarify and substantiate this comment and also how it possibly explains the wider range of fall velocities.

15) Page 6, line 26: I can't find the empirical relationship that is provided in Appendix B.

16) Page 6, line 27: Did the authors mean to say "less humid"? And, is reference being made to the difference in conditions near the surface or aloft in the source/growth region?

17) Page 7, lines 20-22: This sentence needs to be improved for clarity. Does the presence of denser (rimed) particles mean that they have higher terminal velocities, so spend less time in the warmer air and are therefore more likely to succeed in reaching the surface? Reference is made to the wet-bulb temperature, but no actual values are provided. Some readers may also not be familiar with how the wet-bulb temperature affects the evolution of particles below cloud base. A short explanation and citation would be helpful both here and set the stage for discussion that includes the wet-bulb temperature in the subsequent section. The last two sentences at the end of section 4.4 could be moved forward and expanded as part of addressing this.

18) Page 8, lines 8-9: No information is provided on how the melting layer and warm layers changed later in the event, or how these changes relate to the observed changes in precipitation type.

19) Page 8, lines 11-12: I do not think one can claim that the simulations demonstrated the impact of dry conditions on precipitation type reaching the surface, because no simulations were undertaken (or at least included in the text) using relatively moist conditions below cloud base.

20) Page 8, line21: Are the authors referring to 5 dBZ representing a substantial difference between the two profiles of vertical velocity? A return of 5 dBZ is equivalent to a very low precipitation rate.

21) Page 9, lines 4-6: This portion of the text reads awkwardly. I suggest rewording and elaborating for clarity. Also, is there a more important take-away message here about the broader implications of these observations?

Editorial Comments

1) Page 1, line 7: Maybe say, "weather balloons were released" instead of "soundings were launched".

2) Page 1, line 11: Maybe say, "63% showed signs of riming"?

3) Page 1, lines 13-15. Sentence starting with "Radar structure aloft..." lacks clarity and reads awkwardly, I'd suggest rewording.

4) Throughout text: Authors use "m/s". Is this a HESS format? If not, I'd suggest using the conventional ms-1.

5) Page 1, line 17. Suggest replacing "generally" with "relatively".

6) Page 2, line 9: Remove "nonetheless".

7) Page 2, lines 23-24: Suggest removing "...and they can sometimes lead to major disasters such as the 2013 flooding". This has already been mentioned twice before

8) Page 3, lines 9-11: Current wording is awkward. Maybe try "Observations of precipitation type, cloud cover, temperature,10 relative humidity, wind speed, wind direction and surface pressure were typically recorded at 10 minute intervals;..."

9) Page 4, line 15: Please quantify what is meant by "low values of relative humidity".

10) Page 4, line 24: Missing parenthesis after "...Area Model".

11) Page 5, line 2: Wording is not quite right. Specifically, "border northern Alberta".

12) Page 5, lines 6-8: Suggest rewording, "The westerly flow events were generally associated with drier conditions near the surface (Figure 4a) because of adiabatic heating associated from the downslope flow. The easterly flow events produced....".

13) Page 5, line 11: Note sure that "systematically" is the most suitable word here.

14) Page 5, line 16: Replace "events" with "event".

15) Page 5, line 27: Maybe say, "formed in different growth environments"? Are there

any other data that support this claim?

16) Page 5, line32: Suggest removing "...at these temperatures or at higher ones".

17) Page 6, line 8: Remove "solid" before "precipitation".

18) Page 6, line 12: Insert "a". "...associated with a varying flow field".

19) Page 6, line 17: Remove "But".

20) Page 6, line 20. Suggest replacing "...for both types of events" with "...for events during both flow patterns,...".

21) Page 7, line 13: Suggest saying, "Due to the relatively dry..."

22) Page 7, line 14: Suggest, "...been found in this region by Harder and Pomeroy (2013)."

23) Page 7, line 18: Suggest saying, "...at which ice particles..."

24) Page 8, line1: Include the dates and times in parentheses for the rain/mixed precipitation and for the light snow.

25) Page 8, line 1: Suggest following, "These data are interpolated to over 100..."

26) Page 8, line 4: Replace "though" with "through".

27) Page 8, line 18: Make sure that reference is consistently made to either MRR or MRR2 throughout the text.

28) Page 8, line 19: Remove "region".

29) Page 9, line 8: Suggest saying, "...even under relatively dry surface conditions".

30) Page 9, line 25: Suggest following punctuation, "...was observed, or inferred to occur, with many...".

[Figure]

131, 2018.

---

## Referee Comment (RC2) · Anonymous Referee #2 · 16 May 2018

**Review of HESS-2018-131**
**Precipitation characteristics and associated weather conditions on the eastern slopes of the Rocky Mountains during March-April 2015**
**Julie M. Thériault, Ida Hung, Paul Vaquer, Ronald E. Stewart, and John Pomeroy**

This paper presents an analysis of field observations collected in the Canadian Rockies during a 2-month period to better understand how precipitation reaching the surface formed. The authors looked at how precipitation was impacted by flow regimes aloft and could group all their events as being either dominated by westerly or easterly flows. The analysis has a strong focus on the characteristics of solid precipitation. This is a nicely written paper, based on sound data and nicely illustrated.

My main concern is that there is a disconnect between the motivation of the paper (the need to study precipitation extremes in the Rockies) and what the paper is really about. I understand that when you plan a 2-month field campaign, your chances of capturing an extreme event are very small. I also understand that undertaking such a field campaign is particularly demanding and that there is clearly a need for a unique dataset like this one.

Still, I do not understand why out of the 17 events included in the analysis, more than 50% are less than a mm. Actually, 8 of the 17 events are less than 0.2 mm! There is an important fraction of the paper dedicated to the 31 March 2015 event, where a total of 0.03 mm of water equivalent was observed. First, what is the measurement uncertainty of the precipitation gauge? Second, from a hydrological standpoint, what is the interest? Also, on line 8 of p.4, we are told that the cumulative precipitation for the March-April 2015 period is 73 mm. If I sum the precipitation amount of all events listed in Table 1, I get 19.29 mm. Where are the missing 53.71 mm?

I would suggest to remove all the events that are less than 1 mm and to rewrite the introduction (less emphasis on extremes and 2013 flooding) so that I better matches with the actual objectives of the paper.

I have two additional remarks.

p.8, l. 5-6: why is this the case? Explain briefly.

p.9, l. 14-22: Why discuss studies in northern Canada? I would rather focus on other studies looking at precipitation in mountainous terrain.

---

## Author Comment (AC1) · 13 Jun 2018

**Responses to Referee comment #1:**

*The authors would like to thank the reviewer for the constructive comments.*

Overview

The study's authors present a thorough and wide-ranging study of the precipitation characteristics during a field campaign. The manuscript is well written and organized, and the subject material is interesting. The analyses are detailed and robust. The manuscript will, however, benefit by addressing several key concerns identified by the reviewer. My recommendation is publication once the concerns listed below have been appropriately addressed.

Main Concerns

1) The authors state the importance of major summertime floods in the region and that no dedicated atmospheric studies of such events have been made. While true, this claim is problematic when framed in the context of this particular study, which concerns low- intensity precipitation events observed during the spring months. My suggestion would be for the authors to frame the context of this work as it pertains to winter and spring precipitation, and how the characteristics of the observed precipitation/hydrometeor might affect i) the snowpack in terms of SWE, and ii) the spring freshet. On a related note, one could easily justify the need for a similar study in June, when heavy precipitation events typically affect this region.

*The introduction has been reframed with respect to the observed precipitation events document during the project. The revised introduction is as follows.*

*"Western Canada is characterized by complex and rugged terrain where precipitation and associated weather conditions are highly variable (for example Stoelinga et al., 2013). This includes the eastern slopes of the Rocky Mountains, which have a continental climate subject to extremes that fluctuate from severe drought (Hanesiak et al., 2011) to extensive flooding (Pomeroy et al, 2016). Westerly flow from the Pacific Ocean brings heavy precipitation to the west coast of British Columbia while producing dry and warm conditions on these eastern slopes. During the spring, large-scale weather patterns are favourable for easterly, upslope winds that sometimes lead to extreme precipitation events. An example is the flooding in June 2013, which was one of the most catastrophic events in Canadian history (i.e. Liu et al 2016, Kochtubajda et al 2016). Previous floods such as in 2005 (Flesch and Reuter 2012, Shook 2016) rivalled the 2013 event in terms of impact.*

*The type of precipitation reaching the surface on the eastern slopes of the Canadian Rockies varies greatly throughout the cold seasons (Harder and Pomeroy, 2013). As summarized in Liu et al. (2017), the mean annual amount of precipitation (1960-2013) for Banff, Alberta, located ~60 km northwest of Kananaskis, is 61.7 mm. But it is not clear which month receives more precipitation as it is evident that June is the wetter month for Calgary, Alberta (located 100 km east of Kananaskis, Figure 1). Rain-snow transitions at lower elevations generally occur in*

*March and April. Snow-water-equivalent (SWE) at higher elevation reaches a maximum in May and lowers rapidly in June and early July (Pomeroy et al., 2016). For example, catastrophic events such as the 2013 Alberta flooding arose in part because most of the precipitation fell as rain on mountain sides and this acted to melt the existing snowpack and accentuate runoff.*

*The evolution of the snowpack depends strongly on the air temperature as well as the amount and types of precipitation reaching the surface. For the period 1950-2012, winter mean temperatures have increased by 3.9°C (DeBeer et al. 2016) with very little change in precipitation in the eastern slopes of the Canadian Rockies. With the changing climate, it is critical that precipitation be well understood, including its phase, in this area because of its impact in the regional hydrological cycle. In 2008, a field experiment focusing on the changes in precipitation amounts and elevation along a transect perpendicular to the foothills was conducted, the Foothills Orographic Precipitation Experiment (Smith 2008). This experiment defined precipitation-elevation relationship using surface meteorological stations.*

*There is a nonetheless a need to improve our understanding of the atmospheric conditions leading to precipitation as well as the characteristics of the precipitation itself in this area. To address this, a field experiment was carried out in March-April 2015 in the Kananaskis Valley (Figure 1) to obtain critical information such as particle characteristics at the surface as well as atmospheric conditions leading to precipitation events. By utilizing this information, this study aims to better understand the precipitation characteristics and associated atmospheric driving mechanisms on the eastern slopes of the Canadian Rockies during the spring. Some of the specific scientific issues include placing the observing period into a longer-term context, quantifying the temporal variability of precipitation (and its detailed features) at the surface in relation to conditions aloft, and understanding the roles of accretion and sublimation on the precipitation reaching the surface.*

*The manuscript is organized as follows. Section 2 describes the field project and the instrumentation deployed. Section 3 describes the events documented and specific case studies. Section 4 focuses on the characteristics of the precipitation and associated atmospheric conditions and the precipitation processes are presented in Section 5. Section 6 places the results into perspective by comparing its findings with other studies across Canada. Section 6 provides the conclusions."*

2) The authors correctly point out that the climate in this region is changing. However, they do not discuss how the climate has changed in the spring months (i.e., the time period when the observations were made). Also, no connection is made between the long-term trends in temperature and precipitation and the objectives of this study. If they spring climate has been warming and drying, for example, how does that relate to this particular research initiative? That link needs to be clearly stated and substantiated with references to the literature.

*The introduction has been rewritten to reflect the goal of the field campaign. Since that no climate analysis has been done, the climate change information has reorganized. See response to main concern #1.*

3) The reasoning behind the motivation provided on lines 7 through 9 is not clear to me. Mention of made of the importance of precipitation, trends and future occurrence, but no specific discussion on these points is made in the preceding text. The connection between these aspects and this research needs to be clearly laid out.

*This paragraph and the previous one have been re-organized. The full introduction section is included in the main concern #1.*

4) A major focus of this research is how precipitation differs between westerly and easterly flow regimes. In my opinion, while logical, this classification may be too simple, especially considering the location of the study site and change of wind direction with height. For example, oftentimes winds in this region veer markedly with height, so an upslope flow (i.e., easterly) in the low levels may transition to westerly or southwesterly near the barrier height. The absence of winds from the radiosonde soundings, or discussion of vertical wind profiles in general, make correctly differentiating between "true" westerly and easterly events problematic. This could have important implications for the interpretation of results and conclusions drawn in this study. I strongly suggest that the authors use either data from the HRDPS or a higher resolution reanalysis product (e.g., NARR) to generate time-height diagrams of winds above the study site. These data, even with their limitations, can be used to draw more nuanced interpretations of how the winds vary between events and how the vertical profiles of winds affect precipitation.

*The flow field regime has been addressed using the GEM (or HRDPS) outputs as stated on p. 4 line 22 to 25. We had referred to the Appendix A3. We chose the height of 3 km above sea level to correspond to the height of the mountain barrier. As described in Vaquer (2017), for westerly flow events, the wind direction does not significantly change with height but veering is produced at near 3 km above sea level. We considered that when the wind direction had a westerly component it was a westerly flow and a easterly component an easterly flow. We are confident into our analysis as the wind field from 2 km produced the same results.*

*Details on the wind profiles and time series are available in Vaquer (2017). The thesis is available here https://archipel.uqam.ca/10494/1/M15139.pdf. It is in French but you will find the figures of the vertical profile timeseries in Chapter 3 and Annexe A.*

5) Another important consideration is the impact of the local orography on the winds, as this could have important implications on the flow relative to the orography, vertical motions and ultimately precipitation. For example, according to Fig. 1, the study site was located on the western flank of a pronounced ridge on the front range having heights exceeding 2500 m. Whereas an easterly flow would be upslope on the eastern flanks of this ridge, is not clear what happens after the air passes over the ridge-line. It could be that the easterly flow actually results in a localized area of downslope flow on the western flank of the ridge. Similarly, a large-scale

westerly flow could become upslope on the western flank of the ridge, resulting in localized vertical lift.

*Yes, this is a very good point. A comment has been added in the conclusion of the manuscript. "The precipitation at KES could indeed be affected by local topography. Further investigate and very high numerical simulations could be conducted to better understand the link between the larger-scale flow field and the interaction with the orography."*

6) Precipitation amount the manuscript would benefit from a brief discussion of the precipitation amounts observed for the different flow regimes. Also, are heavier amounts typically favoured by a particular flow regime, and if so, why?

*A brief discussion has been added at the end of section 3.2, which is as follows. "As suggested in Table 2, no correlation has been found between the type of flow field regime and the amount of precipitation at the surface. Even if major precipitation events are generally associated with easterly flow field, during this field project, heavier precipitation amounts were not associated with a particular flow field regime. This may be due to a slightly different synoptic scale setup that would had bring less moisture in the area. Further investigation is needed to determine the threshold atmospheric conditions leading to major precipitation events."*

7) The manuscript would also benefit from a discussion on the impact of hydrometeor drift, as non-negligible precipitation amounts can arise because of hydrometeor drift from upslope flow on windward slopes.

*A comment has been added to the conclusion of the manuscript. The following paragraphs also address main concern #5.*

*"Further investigation should be conducted on the local topography effects on precipitation. The local ridge as for the classification of westerly and easterly flow events could have affected the precipitation at KES. Very high numerical simulations could be conducted to better understand the link between the larger-scale flow field and the interaction with the orography. Also, snowdrift was not an issue during the project as we quantified precipitation types and atmospheric conditions up to 6 km in the atmosphere. To investigate, however, snow distribution in complex terrain one needs to take into account that process (e.i. Lehning et al. 2008)."*

8) Data from a sonde mounted on a vehicle (i.e., car-sonde) is referred to in the text. Details concerning these data need to be provide T maybe as an Appendix? Also, reference in the text is made to data gathered whilst ascending Fortress Mountain; please indicate the route and location of Fortress Mountain in Fig. 1.

*A description of the car-sonde was added in an appendix and Fortress Mountain with the road used have been added to Figure 1.*

*Appendix C Description of the car-sonde*
*The car-sonde technique followed the similar methodology as the ski-sonde conducted along Whistler Mountain during the Science of Nowcasting Olympic Weather for Vancouver 2010 (SNOW-V10, Thériault et al., 2014). This technique was used to characterize the height and thickness of the melting layer only when rain was reported in the valley. A hand-held weather station (ex: Kestrel 400) was attached to a stick and held outside the car while ascending the mountain road. At the same time, the exact location was tracked by a GPS and weather conditions, including precipitation types, were recorded as driving. When solid precipitation was reported, microphotography of particles was performed. The route of the car-sonde is on Figure 1c.*

[Figure]

*Figure 1: (a) The Kananaskis Emergency Services (KES) Centre site located in the lee of the Rocky Mountains at 50.9° latitude, -115.1° longitude and (b) is surrounded by topography up to 3010 m ASL. Google Earth. Retrieved on 16 January 2017. (c) is the local topography where the field experiment was held. Fortress Mountain is located south-west of Kananaskis and the route taken for the car-sonde (Appendix C) is indicated in green. The elevations at the base and top of the accessible road are indicated. Retrieved on 30 May 2018.*

Technical Comments

    1) Page 2, lines 13-14: Make sure that this pertains to spring or cold season only.
*It has been clarified.*

    2) Page 2, lines 26-27: The reference to FOPEX seems to be an afterthought. Reasons why the findings from FOPEX are, or are not, relevant to this study need to be clearly substantiated. FOPEX also considered the impact of altitude on precipitation amounts.
*It has been moved to the 3rd paragraph of the introduction. See main concern #1.*

    3) Page 3, line 7: It is not clear what is meant by ". . .taken in a systematic manner". Please reword for clarity.
*The "by scanning the collection pad" has been added to the sentence.*

    4) Page 3 lines 19 and 29: Given the large instrument suite, please make sure that it is clear what instrument is being spoken to in the text. On line 19 are the authors referring to the Geonor gauge? On line 29, what sondes were used? Vaisala RS92s?
*We used a Geonor precipitation gauge and the Vaisala sondes were RS41.*

    5) Page 4, line 3: Is the mean observed RH of 63% really anomalous compared to the average of 65%? Is the difference even statistically significant?
*Relative humidity was not anomalous. It has been corrected.*

    6) Page 4, line 14: Referring to Figure 2, this figure could be another useful contribution from this paper. Although similar, there are noteworthy shifts in the boundaries for these data compared to those from Matsuo et al. There are data points to warrant shifting the demarcation lines between categories. This would be useful to others.
*A comment has been added in the revised version of the manuscript. It is as follows. "There is a shift in the boundaries, in particular the one associated with solid, mixed, and rain. It could be associated the presence of denser particles, which would only melt completely at higher temperatures, compared to wet, non-accreted snow likely occurring in Japan."*

7) Page 4, lines 15-16: The discussion of the wet-bulb temperature comes out of the blue. The importance and relevance of this parameter needs to be introduced first. What do "low values of relative humidity" translate to in percent?

*It has been changed to temperature only for clarity. The concept of wet-bulb temperature is now introduced in section 4.4.*

*8)     Page 4, line 31: This is not a big issue, but using 250-mb to describe synoptic systems is atypical. Was there any particular reason of using data at this level at not 250-mb? Also, it would also be interesting to look at data from 700-mb as this coincides closely to the height of the mountain barrier.*

*Those levels were shown to give an overview of the synoptic scale situation to identify the location of the trough with respect to KES aloft (top of the troposphere) and the surface conditions (SLP map). The SLP map was chosen to show the location of the synoptic scale weather system at the surface along with observations, including wind direction. The 700 hPa maps were added.*

*The paragraph has been improved: "The analyses at 250 hPa, 700 hPa and at sea level are shown in Figure 3. These were chosen to show the large-scale geopotential heights at the top of the troposphere, near the top of the mountain barrier and sea level pressure. The analyses were used to identify the locations of upper-level trough, wind direction, moisture, and surface weather systems with respect to KES. The westerly flow events were generally associated with an upper-level trough located over B.C., which produced large-scale increasing cyclonic vorticity advection with height, favourable for upward motion in the vicinity of KES (Figure 3a). The 700 hPa surface map (Figure 3c) suggests westerly flow with localized high relative humidity (>90%) near KES. The large-scale upper level geopotential heights were supported by the presence of a surface low-pressure system over eastern Alberta (Figure 3e). In contrast, the upper-level trough is located farther west for the westerly flow. This led to a relatively weak ridge near the border of northern Alberta (Figure 3b), which produced subsidence, a high-pressure system and ridge at the surface over western Canada and a weak low-pressure system just south of the Canada-U.S. border (Figure 3f). Higher amounts of moisture (>90%) are available along the eastern slopes of the Canadian Rockies near KES (figure 3d). These were the typical large-scale setups generally leading to westerly and easterly flow field events, respectively."*

[Figure]

Figure 3: (a) and (b) are the 250 hPa analyses of the geopotential heights and wind fields. (c) and (d) are the analysis at 700 hPa that includes relative humidity. The shaded area is relative humidity of >70% and (e) and (f) are the sea level pressure indicated the location of the surface weather systems. (a), (c) and (e) are at 0000 UTC 1 April 2015, which corresponds to a westerly flow event and (b), (d) and (f) are at 0000 UTC 5 April 2015, which corresponds to a easterly flow event.

9) Page 5, line 4: That these synoptic setups corresponded with "all" of the westerly and easterly flow regimes seems too definitive. Unless this can be quantified, maybe more conservative language is warranted.

*The sentence has been reworded to "These were the typical large-scale setups generally leading to westerly and easterly flow field events, respectively.".*

10) Page 5, lines 6-8: This may be pedantic, but the others refer to "dry" several times. It would be more appropriate to say "drier" given that we are referring to precipitation events, albeit with unsaturated conditions below cloud base. Please check text throughout. Also, the wording used to describe the impact of the flow regime on the relative humidity makes it sound like conjecture. Maybe rewording would address this, although using in-situ data that substantiate the claims would be best.

*It has been reworded. The issue of using the "dry" has been addressed throughout the manuscript.*

11) Page 5, line 8: In Fig. 4b the profile looks near-saturated from just above the surface.

*It has been clarified by rewording this paragraph as follows: "To illustrate the atmospheric conditions aloft during both westerly and easterly flow events, a typical example of an atmospheric sounding during each type of event is shown in Figure 4. The westerly flow events were generally associated with drier conditions near the surface (Figure 4a) because of adiabatic heating associated with downslope flow.  It led to near saturation conditions at a similar elevation to that of the barrier east of Kananaskis and near the surface. This easterly flow produced upslope conditions with adiabatic cooling, which led to near saturation conditions throughout the lower troposphere (< 500 hPa, Figure 4b)."*

12) Figures 5 and 6: Panels c and d are swapped in the figure caption. No units are provided for the surface winds, and no explanation of what speeds the barbs represent. It is not clear what sensor1, sensor2 and sensor3 represent please clarify. Is there a particular reason why accumulation estimates from sensor2 are lower than the other sensors for both events?

*The figure caption has been clarified. Panels c and d were switched. An explanation of the wind barb is given in the caption (see Technical comment #13). "The wind barb is like an arrow on which the barb is the wind speed (short line is 5 knots and the long line is 10 knots). It points in the direction to which the wind is blowing" The sensors are the measurement from the Geonor, where 3 different sensors are often used. For clarity, we showed the accumulation averaged over the 3 sensors instead.*

13) Section 3.3. It is not obvious to me why the authors chose to plot the spectral width in Figs. 5,6 and not the Doppler-derived vertical motions. Spectral width data are typically used

as a proxy for turbulent motion. Sound reasoning needs to be provided for the choice of spectral width data when speaking to vertical motions of hydrometeors. The choice of spectral width is also confusing because on line 16 on page 5, reference is made to "the vertical particle motion", which suggests that the authors are referring to radial velocity data.

*The Doppler velocity was plotted instead of the spectral width. The new figures are below.*

[Figure]

*Figure 5: Time evolution (UTC) on 31 March 2015 of (a) reflectivity and (b) vertical particle motion fields retrieved from the Micro Rain Radar 2 (MRR2) for the westerly flow field event. (c) is the surface temperature and relative humidity; (d) is the unadjusted accumulated precipitation at the surface at KES, (e) is the wind speeds and direction represented by the wind barbs, and (f) is the manual observation of precipitation types (MAN) as well as diagnosed using the OTT Parsivel (AUTO) following Battaglia et al (2010). The hatched region on the MRR time series (a, b) indicates ground-level height. The MRR2 quality control was based on Maahn and Kollias (2012). See Appendix A for details on the precipitation type diagnostic method. Note that a wind*

*barb is like an arrow on which the barb is the wind speed (short line is 5 knots and the long line is 10 knots). It points in the direction to which the wind is blowing.*

[Figure]

*Figure 6: Same as Figure 5 but for 4-5 April 2015 for the easterly flow field event.*

14) Page 6, lines 21-22: Regarding ". . .but also due to more regions of upward motions". It is not clear on which data this assertion is made. Please clarify and substantiate this comment and also how it possibly explains the wider range of fall velocities.

*The sentence has been completed with " but also due to more regions of upward motions leading to more riming, which impacts the fallspeed (cf Figure 5,6 and section 5).*

15) Page 6, line 26: I can't find the empirical relationship that is provided in Appendix B.

*They were in the caption of Table 2 but for clarity they were added to the text.*

16) Page 6, line 27: Did the authors mean to say "less humid"? And, is reference being made to the difference in conditions near the surface or aloft in the source/growth region?

*It has been corrected. It refers to the conditions near the surface.*

17) Page 7, lines 20-22: This sentence needs to be improved for clarity. Does the presence of denser (rimed) particles mean that they have higher terminal velocities, so spend less time in the warmer air and are therefore more likely to succeed in reaching the surface? Reference is made to the wet-bulb temperature, but no actual values are provided. Some readers may also not be familiar with how the wet-bulb temperature affects the evolution of particles below cloud base. A short explanation and citation would be helpful both here and set the stage for discussion that includes the wet-bulb temperature in the subsequent section. The last two sentences at the end of section 4.4 could be moved forward and expanded as part of addressing this.

*For clarity, the discussion on wet-bulb temperature has been removed. It has been introduced and explained only in section 4.4. The clarity of section 4.4 has been improved significantly. The reference from Stewart et al. (2015) has been added.*

18) Page 8, lines 8-9: No information is provided on how the melting layer and warm layers changed later in the event, or how these changes relate to the observed changes in precipitation type.

*No temperature feedbacks with the environment have been computed in those simulations to assess the type of precipitation formed within those specific conditions. It has been clarify in the revised version of the manuscript.*

19) Page 8, lines 11-12: I do not think one can claim that the simulations demonstrated the impact of dry conditions on precipitation type reaching the surface, because no simulations were undertaken (or at least included in the text) using relatively moist conditions below cloud base.

*We agree and this claim has been removed from the sentence.*

20) Page 8, line21: Are the authors referring to 5 dBZ representing a substantial difference between the two profiles of vertical velocity? A return of 5 dBZ is equivalent to a very low precipitation rate.

*We agree and the sentence has been reworded as "…, more precipitation (> 5dBz) occurs …".*

21) Page 9, lines 4-6: This portion of the text reads awkwardly. I suggest rewording and elaborating for clarity. Also, is there a more important take-away message here about the broader implications of these observations?

*The text starting at line 4 has been clarified as follows. Note that the first and second statement remained unchanged.*

*"The depth of the layer associated with upward motion is somewhat linked with the degree of riming. During periods with rimed snow at the surface, the upward motion occurred over a shallower layer (~1000 m) than during periods with completely rimed particles (of order ~2000 m). Third, reflectivity values all decreased (except for downslope rain) with descending height just above the surface (~1000 m). This is probably associated with sublimation of particles in that layer. The increase in reflectivity associated with the rain in westerly flow could be produced by the bright band near the surface from the melting of solid precipitation (cf Figure 10).*

*In summary, distinct radar echoes patterns somewhat depended on the type of flow field and the degree of riming of precipitation particles. It was suggested, for example, that growth through accretion occurred in both types of flow field events but at different elevations as well as the importance of sublimation/evaporation in the lower levels of the atmosphere."*

Editorial Comments

1) Page 1, line 7: Maybe say, "weather balloons were released" instead of "soundings were launched".

*It has been changed.*

2) Page 1, line 11: Maybe say, "63% showed signs of riming"?

*It has been changed.*

3) Page 1, lines 13-15. Sentence starting with "Radar structure aloft..." lacks clarity and reads awkwardly, I'd suggest rewording.

*It has been reworded.*

4) Throughout text: Authors use "m/s". Is this a HESS format? If not, I'd suggest using the conventional m s-1.

*It has been modified everywhere.*

5) Page 1, line 17. Suggest replacing "generally" with "relatively".

*It has been changed.*

6) Page 2, line 9: Remove "nonetheless".

*It has been removed.*

7) Page 2, lines 23-24: Suggest removing ". . .and they can sometimes lead to major disasters such as the 2013 flooding". This has already been mentioned twice before

*The first sentence of the paragraph has been changed to "Precipitation events in the Banff/Calgary area can bring rain, snow or both but no atmospheric-oriented special observations (beyond those made with operational networks) have been carried out until the March-April 2015 experiment."*

8) Page 3, lines 9-11: Current wording is awkward. Maybe try "Observations of precipitation type, cloud cover, temperature,10 relative humidity, wind speed, wind direction and surface pressure were typically recorded at 10 minute intervals;. . ."
*It has been reworded.*

9) Page 4, line 15: Please quantify what is meant by "low values of relative humidity".
*It is down to 30%. The value has been added.*

10) Page 4, line 24: Missing parenthesis after "...Area Model".
*It has been added.*

11) Page 5, line 2: Wording is not quite right. Specifically, "border northern Alberta".
*It has been reworded to "northern Alberta border".*

12) Page 5, lines 6-8: Suggest rewording, "The westerly flow events were generally associated with drier conditions near the surface (Figure 4a) because of adiabatic heating associated from the downslope flow. The easterly flow events produced...".
*It has been reworded.*

13) Page 5, line 11: Note sure that "systematically" is the most suitable word here.
*The word systematically has been removed.*

14) Page 5, line 16: Replace "events" with "event".
*It has been changed.*

15) Page 5, line 27: Maybe say, "formed in different growth environments"? Are there any other data that support this claim?
*It has been modified. No other data supported this claim but numerical simulation could eventually help better understanding these growth processes.*

16) Page 5, line 32: Suggest removing "...at these temperatures or at higher ones".
*It has been removed.*

17) Page 6, line 8: Remove "solid" before "precipitation".
*It has been removed.*

18) Page 6, line 12: Insert "a". ". . .associated with a varying flow field".
*It has been added.*

19) Page 6, line 17: Remove "But".
*It has been removed.*

20) Page 6, line 20. Suggest replacing ". . .for both types of events" with ". . .for events during both flow patterns,. . .".
*It has been changed.*

21) Page 7, line 13: Suggest saying, "Due to the relatively dry. . ."
*It has been modified.*

22) Page 7, line 14: Suggest, ". . .been found in this region by Harder and Pomeroy (2013)."
*It has been modified.*

23) Page 7, line 18: Suggest saying, ". . .at which ice particles. . ."
*It has been corrected.*

24) Page 8, line1: Include the dates and times in parentheses for the rain/mixed precipitation and for the light snow.
*The sentence has been reworded for clarity.*

25) Page 8, line 1: Suggest following, "These data are interpolated to over 100. . ."
*It has been changed.*

26) Page 8, line 4: Replace "though" with "through".
*It has been corrected.*

27) Page 8, line 18: Make sure that reference is consistently made to either MRR or MRR2 throughout the text.
*It has been verified and corrected everywhere.*

28) Page 8, line 19: Remove "region".
*It has been removed.*

29) Page 9, line 8: Suggest saying, ". . .even under relatively dry surface conditions".
*It has been changed.*

30) Page 9, line 25: Suggest following punctuation, "...was observed, or inferred to occur, with many...".

*It has been modified.*

---

## Author Comment (AC2) · 13 Jun 2018

**Responses to Referee comment #2 :**

Review of HESS-2018-131 Precipitation characteristics and associated weather conditions on the eastern slopes of the Rocky Mountains during March-April 2015 Julie M. Thériault, Ida Hung, Paul Vaquer, Ronald E. Stewart, and John Pomeroy

*The authors would like to thank the reviwers for the constructive comments.*

This paper presents an analysis of field observations collected in the Canadian Rockies during a 2-month period to better understand how precipitation reaching the surface formed. The authors looked at how precipitation was impacted by flow regimes aloft and could group all their events as being either dominated by westerly or easterly flows. The analysis has a strong focus on the characteristics of solid precipitation. This is a nicely written paper, based on sound data and nicely illustrated.

My main concern is that there is a disconnect between the motivation of the paper (the need to study precipitation extremes in the Rockies) and what the paper is really about. I understand that when you plan a 2-month field campaign, your chances of capturing an extreme event are very small. I also understand that undertaking such a field campaign is particularly demanding and that there is clearly a need for a unique dataset like this one.

*The introduction of the manuscript has been rewritten. It is as follows.*

*"Western Canada is characterized by complex and rugged terrain where precipitation and associated weather conditions are highly variable (for example Stoelinga et al., 2013). This includes the eastern slopes of the Rocky Mountains, which have a continental climate subject to extremes that fluctuate from severe drought (Hanesiak et al., 2011) to extensive flooding (Pomeroy et al, 2016). Westerly flow from the Pacific Ocean brings heavy precipitation to the west coast of British Columbia while producing dry and warm conditions on these eastern slopes. During the spring, large-scale weather patterns are favourable for easterly, upslope winds that sometimes lead to extreme precipitation events. An example is the flooding in June 2013, which was one of the most catastrophic events in Canadian history (i.e. Liu et al 2016, Kochtubajda et al 2016). Previous floods such as in 2005 (Flesch and Reuter 2012, Shook 2016) rivalled the 2013 event in terms of impact.*

*The type of precipitation reaching the surface on the eastern slopes of the Canadian Rockies varies greatly throughout the cold seasons (Harder and Pomeroy, 2013). As summarized in Liu et al. (2017), the mean annual amount of precipitation (1960-2013) for Banff, Alberta, located ~60 km northwest of Kananaskis, is 61.7 mm. But it is not clear which month receives more precipitation as it is evident that June is the wetter month for Calgary, Alberta (located 100 km east of Kananaskis, Figure 1). Rain-snow transitions at lower elevations generally occur in March and April. Snow-water-equivalent (SWE) at higher elevation reaches a maximum in May and lowers rapidly in June and early July (Pomeroy et al., 2016). For example, catastrophic*

*events such as the 2013 Alberta flooding arose in part because most of the precipitation fell as rain on mountain sides and this acted to melt the existing snowpack and accentuate runoff.*

*The evolution of the snowpack depends strongly on the air temperature as well as the amount and types of precipitation reaching the surface. For the period 1950-2012, winter mean temperatures have increased by 3.9°C (DeBeer et al. 2016) with very little change in precipitation in the eastern slopes of the Canadian Rockies. With the changing climate, it is critical that precipitation be well understood, including its phase, in this area because of its impact in the regional hydrological cycle. In 2008, a field experiment focusing on the changes in precipitation amounts and elevation along a transect perpendicular to the foothills was conducted, the Foothills Orographic Precipitation Experiment (Smith 2008). This experiment defined precipitation-elevation relationship using surface meteorological stations.*

*There is a nonetheless a need to improve our understanding of the atmospheric conditions leading to precipitation as well as the characteristics of the precipitation itself in this area. To address this, a field experiment was carried out in March-April 2015 in the Kananaskis Valley (Figure 1) to obtain critical information such as particle characteristics at the surface as well as atmospheric conditions leading to precipitation events. By utilizing this information, this study aims to better understand the precipitation characteristics and associated atmospheric driving mechanisms on the eastern slopes of the Canadian Rockies during the spring. Some of the specific scientific issues include placing the observing period into a longer-term context, quantifying the temporal variability of precipitation (and its detailed features) at the surface in relation to conditions aloft, and understanding the roles of accretion and sublimation on the precipitation reaching the surface.*

*The manuscript is organized as follows. Section 2 describes the field project and the instrumentation deployed. Section 3 describes the events documented and specific case studies. Section 4 focuses on the characteristics of the precipitation and associated atmospheric conditions and the precipitation processes are presented in Section 5. Section 6 places the results into perspective by comparing its findings with other studies across Canada. Section 6 provides the conclusions."*

Still, I do not understand why out of the 17 events included in the analysis, more than 50% are less than a mm. Actually, 8 of the 17 events are less than 0.2 mm! There is an important fraction of the paper dedicated to the 31 March 2015 event, where a total of 0.03 mm of water equivalent was observed.

*Mistakes were found in the accumulated precipitation shown in Table 1. A revised version has been made. The authors are sorry about this and we really appreciate the reviewer finding this error. The accumulated precipitation at Hay Meadow was used for consistency. When less than 0.2 mm was recorded, it is indicated as a trace. The revised Table 1 is as follows.*

*Table 1: A summary of the events and instrumentation over the March-April 2015 observing period at the KES site. For each event, the mean temperature (T) and relative humidity (RH) are given. The amount and types of precipitation (Amount pcpn) at the surface were rain (R), mixed*

(M) and/or snow (S). The observed type was at KES and the amounts shown are at Hay Meadow (HM, Marmot creek). The amount of precipitation was adjusted using Smith (2007). A trace is when accumulated precipitation is <0.2 mm. The direction of the flow field aloft (Type event), which was either an easterly flow event (E) or a westerly event (W), is also indicated. A X indicates when each instrument was operating. These are the surface manual observations (SMO), the surface meteorological station (SMS), the optical disdrometer (OD), the Micro Rain Radar (MRR2), the microphotography setup (MP) and the sounding system (SS).

| Num | Event | $\bar{T}$ [°] | $\bar{RH}$ [%] | Amount pcpn [mm] | Type pcpn | Duration [h] | Type event | SMO | SMS | OD | MRR2 | MP | SS |
|---|---|---|---|---|---|---|---|---|---|---|---|---|---|
| 1 | 15-16 March | | | 11.3 | S | 21 | E | | | X | | X | |
| 2 | 21-22 March | | | 1.9 | R→M→S | 11 | W | X | X | X | | X | |
| 3 | 23 March | | | 1.2 | S→M | 3 | E | X | X | | X | | X |
| 4 | 28 March | 5.7 | 82 | 4.0 | R→M | 6 | W | X | X | X | X | | X |
| 5 | 30 March | 6.1 | 67 | trace | R | 2 | W | X | X | | X | | X |
| 6 | 31 March | 7.0 | 58 | 2.0 | R→M | 3 | W | X | X | X | X | | X |
| 7 | 1-2 April | 0.8 | 54 | 4.4 | S | 9 | E | X | X | X | X | | X |
| 8 | 4-5 April | -0.5 | 83 | 3.0 | S | 15 | E | X | X | X | X | X | X |
| 9 | 6 April | -0.5 | 62 | 0.3 | S | 6 | W | X | X | X | X | | X |
| 10 | 11-12 April | 2 | 63 | 2.2 | S | 20 | W | X | X | X | X | X | X |
| 11 | 14-15 April | 3.2 | 76 | 7.9 | R→M→S | 11 | W | X | X | X | X | X | X |
| 12 | 17 April | 12.4 | 52 | 0.4 | R | 2 | W | X | X | X | X | | X |
| 13 | 18 April | 1.6 | 79 | 5.1 | R→M→S | 16.75 | W | X | X | X | X | X | X |
| 14 | 22-23 April | 9.9 | 42 | 1.1 | R→S | 7 | W | X | X | X | X | | X |
| 15 | 24-25 April | 6.3 | 49 | 1.6 | S | 11 | W | X | X | X | X | | X |
| 16 | 25-26 April | 3.4 | 65 | 0.6 | M→S | 6 | E | X | X | X | X | X | X |
| 17 | 29 April | 10.6 | 46 | 1.4 | R→M→S | 10 | W | X | X | X | X | | X |

First, what is the measurement uncertainty of the precipitation gauge?

*The uncertainty on the precipitation gauge is relatively low because wind speed was less than 5 m/s and most of the solid precipitation was rimed particles, which has a better collection efficiency than dry snow (Thériault et al., 2012). Also, wind speed and directions at KES were not at standard heights. For clarity, we, decided to use the precipitation measured at Hay Meadow, which was adjusted for wind speed following Smith (2007). The precipitation distribution and timing is different than KES but the data are consistent among them.*

Second, from a hydrological standpoint, what is the interest?

*The interest from an hydrological point of view is to quantify the type of precipitation to know if the particles y are solid, liquid or mixed phase. This will impact their interaction with the surface*

*and determine, for example, if the precipitation runs off directly into rivers directly or accumulates onto the snowpack.*

Also, on line 8 of p.4, we are told that the cumulative precipitation for the March-April 2015 period is 73 mm. If I sum the precipitation amount of all events listed in Table 1, I get 19.29 mm. Where are the missing 53.71 mm?

*There was a mistake in the data. The cumulative precipitation at Hay Meadow is 68.9 mm. During our project (15 march to April 30, 2015), we measured documented events that produced 47.4 mm at Hay Meadow. We missed precipitation events from 1-14 March 2015. Note also that precipitation accumulation may be different at KES but Hay Meadow give a good indication of when the precipitation occurred.*

I would suggest to remove all the events that are less than 1 mm and to rewrite the introduction (less emphasis on extremes and 2013 flooding) so that I better matches with the actual objectives of the paper.

*The authors had considered removing events with less than 1 mm of precipitation. We decided, however, to keep all events in the analysis because the focus is on characterizing precipitation reaching the surface. If the instruments captured some particles and the observed reported precipitation, we need to consider them.*

I have two additional remarks. p.8, l. 5-6: why is this the case? Explain briefly.

*When non-accreted snow falls through the atmosphere, it melts faster than rimed snow. It will then produce only rain, whereas snow pellets can melt partially producing a combination of precipitation types (rain and mixed phase particles, for example).*

p.9, l. 14-22: Why discuss studies in northern Canada? I would rather focus on other studies looking at precipitation in mountainous terrain.

*We discussed atmospheric related studies that also examined instances of low precipitation rates at the surface. It happened that these were from northern Canada although most were also affected by strong topography. A sentence in the first paragraph has been added.*

*"Most of the solid particles found during March-April 2015 were rimed or were mixed with unrimed particles even under relatively dry surface conditions. These findings related to precipitation events associated with low intensity and rimed snow can be compared to the findings of some previous studies in other regions experiencing similar cold season precipitation."*

*A paragraph on other studies in complex terrain has been added. The following paragraph would the fourth paragraph of section 6.*

*"Other field projects were also conducted over complex terrain (i.e. MAP, Steiner et al, 2003; OLYMPEX, Houze et al. 2017) but few focused on the type of precipitation, in particular, the characteristics of solid precipitation. It is possible however to document the degree or riming automatically using a Multi-Angle snowflake camera (MASC, Garrett et al, 2012) as proposed, by Praz et al. (2017). They documented solid precipitation and the associated degree of riming in Antarctica."*